# A theory of hippocampal theta correlations accounting for extrinsic and intrinsic sequences

Yuk-Hoi Yiu[1,2], Christian Leibold[1,3]*

[1]Fakultät für Biologie & Bernstein Center Freiburg Albert-Ludwigs-Universität Freiburg, Freiburg, Germany; [2]Graduate School of Systemic Neurosciences, Ludwig-Maximilians-Universität München, Munich, Germany; [3]BrainLinks-BrainTools, Albert-Ludwigs-Universität Freiburg, Freiburg, Germany

*For correspondence:
christian.leibold@biologie.uni-freiburg.de

Competing interest: The authors declare that no competing interests exist.

**Abstract** Hippocampal place cell sequences have been hypothesized to serve as diverse purposes as the induction of synaptic plasticity, formation and consolidation of long-term memories, or navigation and planning. During spatial behaviors of rodents, sequential firing of place cells at the theta timescale (known as theta sequences) encodes running trajectories, which can be considered as one-dimensional behavioral sequences of traversed locations. In a two-dimensional space, however, each single location can be visited along arbitrary one-dimensional running trajectories. Thus, a place cell will generally take part in multiple different theta sequences, raising questions about how this two-dimensional topology can be reconciled with the idea of hippocampal sequences underlying memory of (one-dimensional) episodes. Here, we propose a computational model of cornu ammonis 3 (CA3) and dentate gyrus (DG), where sensorimotor input drives the direction-dependent (extrinsic) theta sequences within CA3 reflecting the two-dimensional spatial topology, whereas the intrahippocampal CA3-DG projections concurrently produce intrinsic sequences that are independent of the specific running trajectory. Consistent with experimental data, intrinsic theta sequences are less prominent, but can nevertheless be detected during theta activity, thereby serving as running-direction independent landmark cues. We hypothesize that the intrinsic sequences largely reflect replay and preplay activity during non-theta states.

## eLife assessment

This **important** work presents an interesting perspective for the generation and interpretation of phase precession in the hippocampal formation. Through numerical simulations and comparison to experiments, the study provides a **convincing** theoretical framework explaining the segregation of sequences reflecting navigation and sequences reflecting internal dynamics in the DG-CA3 loop. This study will be of interest for researchers in the spatial navigation and computational neuroscience fields.

## Introduction

As a rat navigates in an environment, place cells fire sequentially during one theta cycle (~100 ms) and form time-compressed representations of behavioral experiences (*Skaggs et al., 1996*), called theta sequences. Theta sequences were proposed to be driven by extrinsic (extrahippocampal) sensorimotor input (*Foster and Wilson, 2007*; *Huxter et al., 2008*; *Romani and Tsodyks, 2015*; *Yiu et al., 2022*), since they are played out in the direction of travel during locomotion and, hence, represent current behavioral trajectories. In contrast, various types of hippocampal sequences have also been

proposed to arise from intrinsic hippocampal connectivity. Non-local activation of place sequences during immobile periods was observed in replay of past locations after the space has been explored (*Skaggs and McNaughton, 1996*; *Lee and Wilson, 2002*) as well as in preplay (*Dragoi and Tonegawa, 2011*) of prospective locations before the animal explores a novel environment. In addition, some CA3 place cells exhibit out-of-field firing at reward locations (*Sasaki et al., 2018*). These remote activations of place cells reflect the underlying circuit connectivity rather than the actual location and movement of the animal. Furthermore, a subset of CA3 cell pairs shows rigid theta correlations with peak lags that are independent of the traversal order of their place fields (*Yiu et al., 2022*), suggesting the existence of hard-wired sequences *even* when sensorimotor drive is present. Such intrinsic sequences that are driven by intrahippocampal connectivity (*Tsodyks et al., 1996*), although less predominantly observed during theta (*Yiu et al., 2022*), are generally interpreted as reflecting spatial memories or planning (*Kay et al., 2020*).

Existing models for theta sequences are, however, either fully extrinsic or intrinsic. The former often employ short-term plasticity (*Romani and Tsodyks, 2015*; *Thurley et al., 2008*), which creates synaptic couplings that are temporally stronger along the instantaneous forward direction. In contrast, intrinsic models, such as the *Tsodyks et al., 1996* model, use a fixed asymmetrical weight matrix pre-designed to align with one movement trajectory (for review see *Maurer and McNaughton, 2007*; *Jaramillo and Kempter, 2017*). Neither of these models alone can explain the simultaneous presence of rigid and flexible correlations in theta sequences. Here we present a network model that accounts for both types of correlations by separating their generation into two anatomically distinct layers: CA3 and dentate gyrus (DG). Extrinsic sequences are generated in the CA3 layer by short-term synaptic plasticity mechanisms, while the intrinsic sequences are evoked by the CA3-DG feedback loop with fixed asymmetrical weights, as inspired by the experimental evidence that lesions of DG abolish non-local activation of CA3 place cells (*Sasaki et al., 2018*) and CA3 theta correlations (*Ahmadi et al., 2022*).

In this paper, we present a model for theta correlations that unifies both extrinsic and intrinsic mechanisms. Extrinsic and intrinsic sequences can propagate simultaneously in separate directions, along the movement trajectory and the pre-designed CA3-DG feedback loops, respectively. As a result, spike correlations display directionality as the two sequences cross each other at various angles: The more parallel they are, the stronger the correlation. Our simulations are in quantitative agreement with directionality properties found in experimental data (*Yiu et al., 2022*) and propose that rigid correlation structure can serve as a stable temporal pattern, which is recognizable across multiple movement directions. This temporal 'landmark' pattern allows spatial encoding even if sensory-motor experience is lacking and may reflect the mechanistic basis for replay in non-theta states.

## Results
### Dependence of theta sequences on heading directions: Extrinsic and intrinsic sequences

Theta-scale correlations of place cells have been explained by previous models using two different types of network mechanisms, intrinsic and extrinsic ones. For *intrinsic* models spike correlations are explained by only the recurrent connectivity of the neuronal network (*Figure 1A*). For *extrinsic* models, the spike correlation is defined by sensory-motor parameters such as movements (*Figure 1B*). We first illustrate how these mechanisms work for two exemplary representatives of these two major model classes.

For intrinsic models, we refer to the *Tsodyks et al., 1996* model where phase precession is generated by the fixed asymmetrical connectivity between place cells. Spike phases of the place cells ahead of the animal decrease as the excitatory drive is gradually increasing, but only along the direction in which the connection strength is asymmetrically stronger (e.g. rightward in *Figure 1A*), called the *asymmetry direction*. Here we simulate a network of CA3 place cells with fixed asymmetrical connectivity (see Methods section: CA3 recurrent connections) as suggested in the *Tsodyks et al., 1996* model and applied our model to behavioral running trajectories in a 2-d open space (*Figure 1C*). Phase precession and spike correlations (*Figure 1D–E*) are compared for opposite running directions. In our simulations, all place cells project excitatory synapses to their counterparts with rightward neighboring place fields. Phase precession therefore is determined by how closely the running direction matches

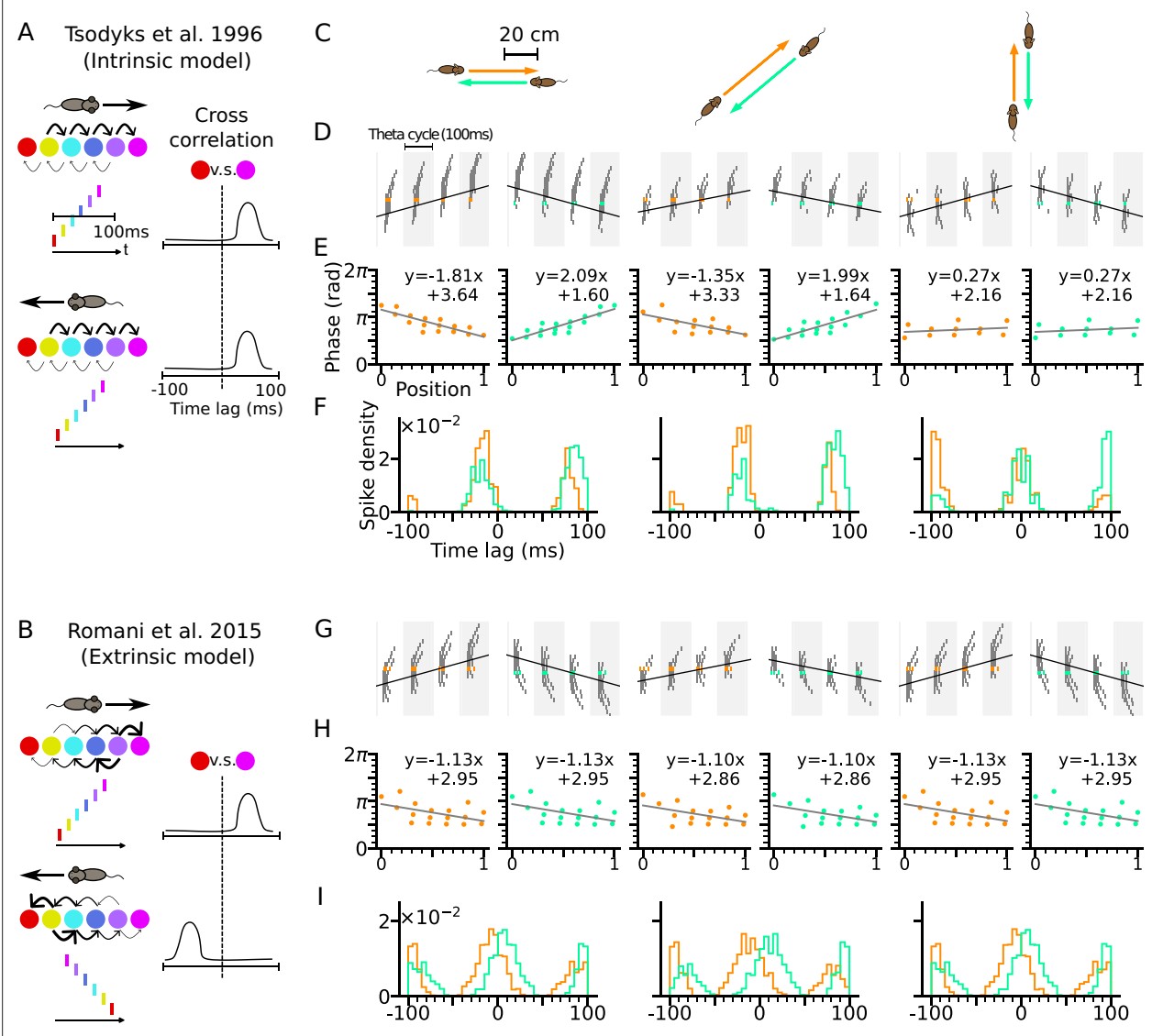

**Figure 1.** Phase precession depends on running direction in intrinsic models but not extrinsic models. (**A**) Left: Schematic illustration of the intrinsic *Tsodyks et al., 1996* model. When the animal runs through a series of place fields (solid circles) in 1-D, the place cells fire action potentials in sequence at the theta timescale (spike raster with corresponding colors). Recurrent connectivity is pre-configured and asymmetrical (connection strengths indicated by arrow sizes). Right: Cross-correlation function between the red and magenta cell, which remains the same for both running directions. (**B**) Left: Schematic illustration of the extrinsic *Romani and Tsodyks, 2015* model. Recurrent connections behind the animal are temporarily depressed by short-term plasticity, and thereby, become movement-dependent. Right: Cross-correlation flips sign in the opposite running direction. (**C**) Simulated trajectories (duration 2 s) in a 2-d environment (80×80 cm) with speed 20 cm/s in left and right (left column), diagonal (middle), and up and down (right) directions. (**D–F**) Simulation results from the intrinsic model (with fixed asymmetrical connectivity inspired by the *Tsodyks et al., 1996* model). Place cells only project synapses to their right neighbors. (**D**) Spike raster plots of place cells along the orange (left panel) and light green (right panel) trajectories (colors defined in C). Theta sequence order remains the same in the reversed running direction. Black line indicates animal position. (**E**) Phase-position relation for the spikes colored in C. Linear-circular regression (gray line) parameters are indicated on top. Positions of the animal at the first and last spike are normalized to 0 and 1, respectively. (**F**) Averaged cross-correlation of all cell pairs separated by 4 cm along the trajectory. Reversal of running direction does not flip the sign of the peak lags. (**G–I**) Same as D-F, but for the extrinsic model (spike-based variant of *Romani and Tsodyks, 2015* model). Correlation peaks flip after reversal of running direction.

the asymmetry direction imposed by the intrinsic connections. The closer the trajectory angle aligns with this asymmetry direction, the more negative is the slope of phase precession (*Figure 1E*). Since in this case, the theta sequence only propagates rightwards as place cells are sequentially activated from left to right, the signs of spike correlations between cell pairs remain invariant to the movement direction (*Figure 1F*, see Methods section: Cross-correlation analysis). Intrinsic models thus cannot explain

experimentally observed directional independence of phase precession and directional dependence of theta spike correlations (*Huxter et al., 2008*).

Our example of an extrinsic model is based on our spiking simulations of the originally rate-based model by *Romani and Tsodyks, 2015*, where phase precession was explained by symmetric recurrent connections that undergo running direction-dependent attenuation by short-term synaptic depression (STD): place fields with centers behind the current animal position on the trajectory thereby received largely reduced recurrent input resulting in recurrently driven theta sequences to play out only in forward direction (see Methods section: CA3 recurrent connections). We simulated our spiking variant of the *Romani and Tsodyks, 2015* model with the same trajectories as the intrinsic model (*Figure 1G–I*), and recovered direction-independent phase precession (*Figure 1H*). Since now, the theta sequences are played out in the same direction as the movement, theta spike correlations are symmetrically reversed (*Figure 1I*) as shown experimentally in CA1 neurons (*Huxter et al., 2008*; *Yiu et al., 2022*).

In area CA3, however, theta spike correlations are neither solely extrinsic (*Yiu et al., 2022*; *Kay et al., 2020*), since phase precession properties change in relation to running directions, nor are they solely intrinsic since reversal of correlation is still observed in most of the sequences (*Huxter et al., 2008*; *Yiu et al., 2022*). We therefore propose a new theory of phase precession for CA3 incorporating both intrinsic and extrinsic factors.

## Directional sensory input

To, however, fully explain directional properties of theta phase precession and theta spike correlations by a model, also directional modulations of firing rates (*Yiu et al., 2022*) need to be taken into account.

We therefore included both directional and positional modulation of the sensory input to the model place cells (*Figure 2A–B*) with randomized preferred heading directions (see Methods section: Spatial input). The sensory input is assumed to arise from MEC, and hence, it is also theta-modulated and phase-shifted by 70° with respected to the peak of theta cycle (*Mizuseki et al., 2009*). Furthermore, since the precession slope observed in the *Romani and Tsodyks, 2015* model is limited (-1.13 radians per field size, see *Figure 1H*) as compared to the experimental reports (-4.44 radians (*Yiu et al., 2022*) and about -2.0 radians (*Harris et al., 2002*) per field size), we introduced short-term synaptic facilitation (STF) to the sensory input (*Berretta and Jones, 1996*; *Thurley et al., 2008*) generating temporally asymmetric depolarization as suggested by intracellular recordings in vivo (*Harvey et al., 2009*; *Figure 2B*). STF amplifies the sensory current at the later part of the field, thus creating phase precession with steeper slopes thereby extending the phase range (see Methods section: Spatial input). Finally, we designated the synaptic weights to be stronger between place cells with similar preferred heading directions (*Figure 2C*) as has been proposed (*Brunel and Trullier, 1998*) as a result of Hebbian plasticity applied to directional firing fields.

A simulation of the place cell network was performed for a rightward trajectory through the arena based on our variant of the extrinsic *Romani and Tsodyks, 2015* model (*Figure 2D*). We focus on two sets of place cells, one for which the trajectory aligns with the preferred heading direction of the field (red, denoted as *best* direction) and one for which the trajectory runs opposite the preferred heading direction (denoted as *worst* direction; *Figure 2F*). Phase precession has a lower onset and marginal spike phase along best direction than along the worst (*Figure 2G–H*), which is consistent with experimental data (*Yiu et al., 2022* report mean spike phases ± SEM for best and worst direction of 1.61 ± 0.02 and 2.22 ± 0.03 in radians respectively), reflects that larger depolarizations generally yield shorter latencies. Directionality of the input, although it yields lower spike phases through higher depolarization, does not affect spike pair correlations, which remains solely extrinsic (*Figure 2I*). Thus, even though rate directionality and directional bias in recurrent connectivity can render phase precession directionally dependent, they are not sufficient to account for intrinsic sequences.

## Generation of intrinsic sequences by the DG-CA3 recurrent network

To explain the expression of intrinsic sequences in CA3, we propose them to be generated by the interaction of two networks, CA3 and DG (*Figure 3A*). DG is a good candidate region to be involved in phase precession, since lesions of it were shown to reduce prospective spiking (*Sasaki et al., 2018*) and to lower the onset phase of phase precession (*Ahmadi et al., 2022*). In our model, the neurons in

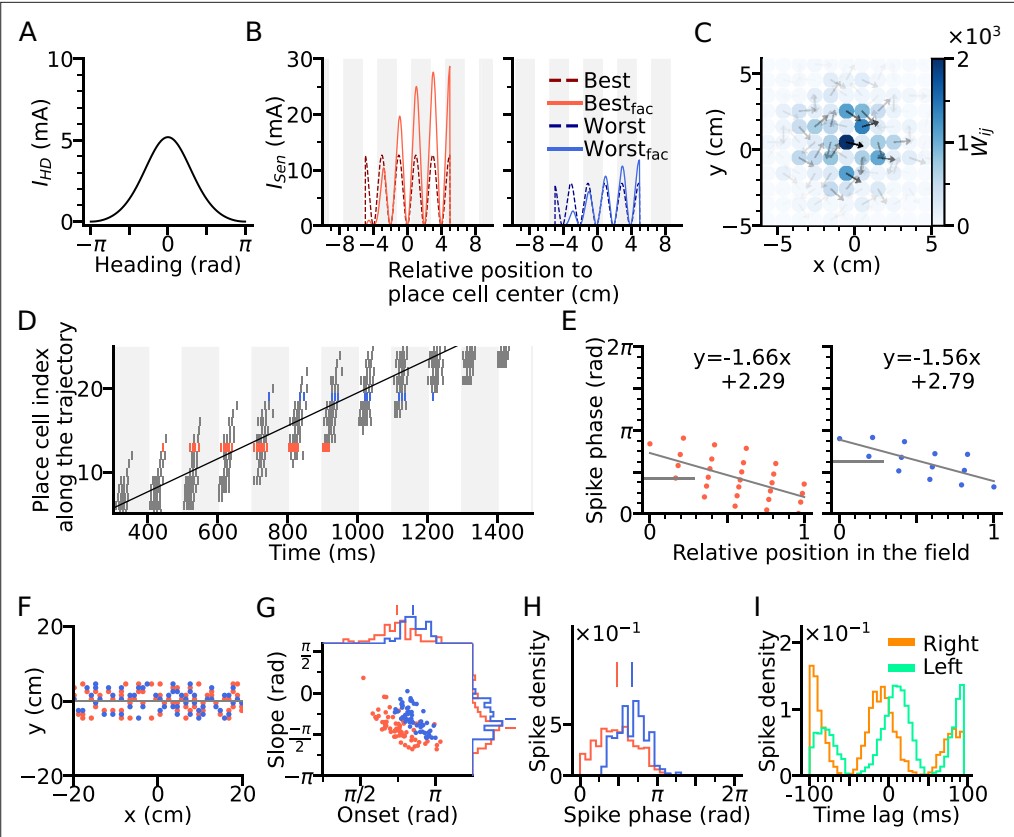

**Figure 2.** Directional input gives rise to spikes at lower theta phase. (**A**) Directional input component of an example place cell. (**B**) Total sensory input as the sum of directional and positional drive of an example place cell for the animal running along (red dashed line, left) and opposite (blue dashed line, right) the preferred heading direction of the cell, respectively named as *best* and *worst* direction. The sensory input is modelled by oscillatory currents arriving with +70° phase shift relative to theta peaks (gray vertical lines). Place fields are defined by a 5 cm rectangular envelope. Solid lines depict the input current including short-term synaptic facilitation. (**C**) Synaptic weights ($W_{ij}$, color) from the place cell at the center (the darkest dot) to its neighbors in the 2-d environment. Each dot is a place field center in 2-d space. Arrows depict their preferred heading directions. (**D**) Spike raster plot sorted by visiting order of the place fields along the trajectory. Spikes of the cells with best and worst direction are colored in red and blue, respectively. (**E**) Phase position plots for the cells with best and worst direction from D (labels as in **Figure 1E**). The mean phase is marked as horizontal gray bar. (**F**) Example place cell centers with best (<30° different from the trajectory; red) and worst (>150°; blue) directions relative to the rightward trajectory (gray line). Only centers of cells that fire more than 5 spikes are shown. (**G**) Slopes and onsets of phase precession of the population from (**F**). Marginal slope and onset distributions are plotted on top and right, respectively. Note higher phase onset in the worst-direction case. (**H**) Spike phase distributions. Higher directional inputs generate lower spike phases. (**I**) Average spike correlation between all pairs with 4 cm of horizontal distance difference when the animal runs rightwards and leftwards. Peak lags are flipped as expected from an extrinsic model.

DG receive excitatory synaptic inputs from CA3 place cells (putatively via hilar mossy cells) and project back to the CA3 cells with place field centers at a different location (**Equation 3**) to induce propagation of intrinsic sequences along a specific spatial direction. The CA3 cells at the target location of DG input are then activated and evoke higher depolarization in cells with place fields at the next DG target locations through the feedback. This scheme produces a rigid sequence whose activation order is independent of the movement direction. The connection pattern of DG-CA3 projections (for brevity, we also refer to it as 'DG loop' in the subsequent text) could be determined by pre-existing network structure or past experience through associative learning, or both.

 **Figure 3**, provides schematic illustrations, for a DG layer that either only projects CA3 activity to their rightward neighbours ($\theta_{DG} = 0°$, **Figure 3A**) or only to their leftward neighbors ($\theta_{DG} = 180°$, **Figure 3E**). Simulations for both cases ($\theta_{DG} = 0°$ and $\theta_{DG} = 180°$) assume a rightward trajectory. Apart

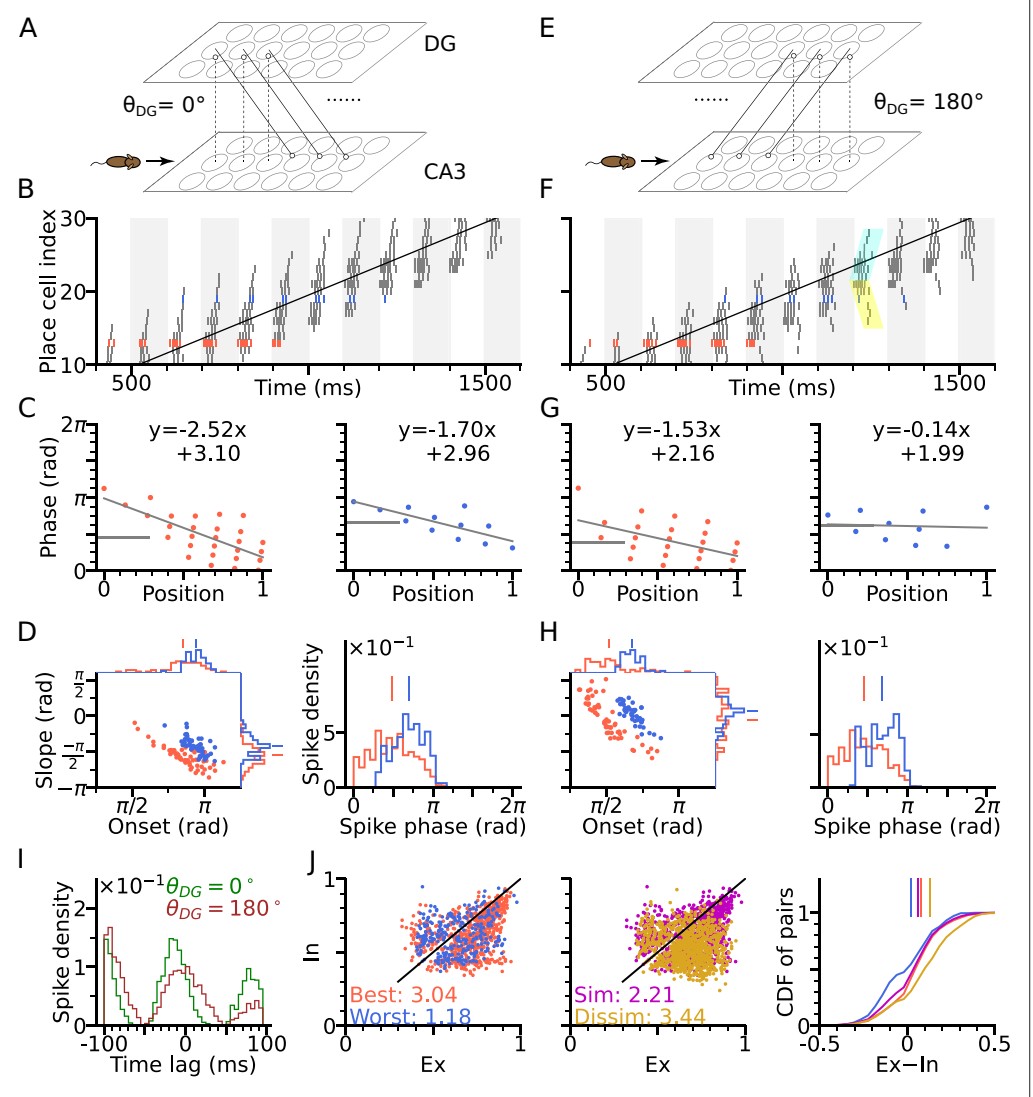

**Figure 3.** DG-CA3 loop introduces directionality of theta sequences. (**A**) Illustration of synaptic connections from CA3 place cells to DG and vice versa. DG layer mirrors the place cell population in CA3 and redirects the CA3 inputs back to different locations. Here, DG cells project into CA3 place cells with fields 4 cm displaced to the right of the pre-synaptic CA3 cells. $\theta_{DG}$ denotes the angular difference between the DG projection direction and the animal's movement direction. (**B**) Spike raster plots sorted by cell indices along the trajectory (2 s duration) from x=-20 cm to x=20 cm. Cells with best and worst angles are marked by red and blue colors, respectively. (**C**) Phase-position plots as is **Figure 2E**. (**D**) Distributions of precession slopes, onsets and spike phases as in **Figure 2G–H**. (**E–H**) Same as A-D, but with DG cells projecting opposite to the animal's movement direction ($\theta_{DG} = 180°$). In F, cyan and yellow shaded regions indicate the examples of forward sequence induced by the movement (extrinsic), and backward sequence induced by the DG recurrence (intrinsic), respectively. (**I**) Average spike correlations for $\theta_{DG} = 0°$ and $\theta_{DG} = 180°$ for pairs separated by 4 cm along the trajectory. Note that for $\theta_{DG} = 180°$, there is a relative excess of spike-pairs with positive lags. (**J**) Left: Intrinsicity and extrinsicity (see Methods) for all pairs from the populations with best (red) and worst (blue) direction. Pair correlations above and below the identity line are classified as intrinsic and extrinsic, respectively. Numbers are the ratios of extrinsically to intrinsically correlated field pairs. Note that the red best direction pairs are more extrinsic than the blue worst direction pairs due to higher sensory input. Middle: Ex/Intrinsicity of pairs with similar (<30°) and dissimilar (>150°) preferred heading angles. Pairs with similar preferred heading angle s are more intrinsic due to stronger DG-CA3 recurrence. Right: Cumulative distribution of the differences between extrinsicity and intrinsicity. Dissimilar and best direction pairs have higher bias to extrinsicity than similar and worst direction pairs, respectively.

The online version of this article includes the following figure supplement(s) for figure 3:

*Figure 3 continued on next page*

*Figure 3 continued*

**Figure supplement 1.** Effects of STF and STD time constants on theta sequences.

**Figure supplement 2.** Effects of running speed on theta sequences.

from the addition of the DG layer, the model architecture and parameters of CA3 layer are the same as in *Figure 2* (including best and worst direction in place field firing rate), which only generates extrinsic sequences through STD in the CA3 recurrent synaptic connections. DG-loop connectivity is additionally modulated by firing rate directionality of the CA3 place fields. Fields with similar preferred heading directions are more strongly connected via the loop than those with opposite preferred heading directions (see Methods section: DG layer).

We found that, when the simulated animal is running in the same direction as the DG-CA3 projection, phase precession starts from a higher phase (*Figure 3C–D*) due to the forward activation of place cells through DG layer (recovering the effect of asymmetric connectivity in the original *Tsodyks et al., 1996* model), as compared to the model without DG layer (*Figure 2G–H*). Spike phases in best direction remain lower than along the worst direction (*Figure 3D*). When, however, the animal is running against the DG-CA3 projection (*Figure 3E*), extrinsic sequences are still present in forward direction, evoked by the movement of the animal, but the intrinsic sequences are played out backward as determined by the direction of fixed recurrence (see cyan and yellow shaded regions in *Figure 3F*). The latter is reflected by the higher phase at the end of the phase position plots (*Figure 3G*) which leads to flatter precession slopes and decreases the fraction of phase precession (slope <0) of all traversal trials (*Figure 3H*). A closer look into pair correlation reveals that for trajectories opposite to the DG-loop projection ($\theta_{DG} = 180°$), spike probability is added to positive time lags (*Figure 3I*). Therefore, introducing fixed recurrence through DG loops elicits both extrinsic and intrinsic sequences and qualitatively changes theta sequences.

To quantify the degrees of extrinsic and intrinsic sequence firing in a way allowing comparison to experimental reports, we use the measures *extrinsicity* and *intrinsicity* (*Yiu et al., 2022*) that are based on pairs of place cells with overlapping place fields (see Methods section: Extrinsicity and intrinsicity, and Discussion). In our simulation, extrinsically and intrinsically driven cell pairs are both present in the population (*Figure 3J*), indicating a coexistence of extrinsic and intrinsic sequences. Our model reproduces a greater extrinsicity for cell pair activity when running direction aligns with both best place field directions as compared to when it aligns to both worst field directions, since along the best direction,

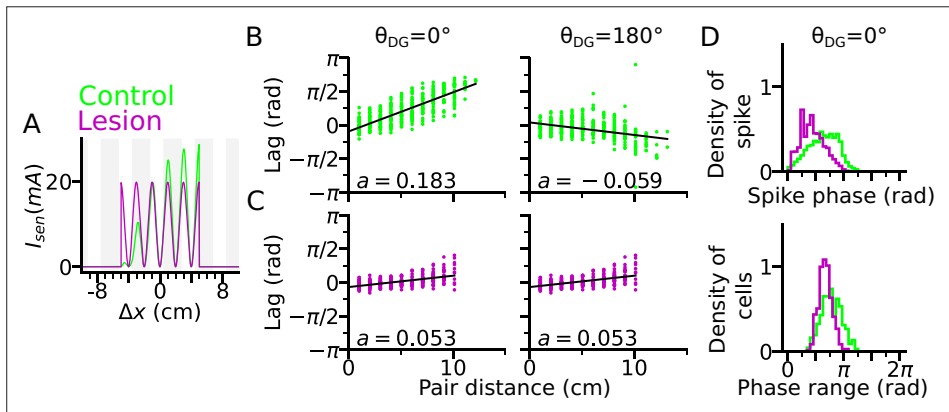

**Figure 4.** DG lesion reduces theta compression and phase precession range. DG recurrence is turned off to simulate the lesion condition. (**A**) Positional sensory inputs into a place cell in lesion (purple) and control (green) cases. The control case is identical to *Figure 3*. In the lesion case, DG input is compensated by increased sensory input with increased probability of synaptic release, hence reduced short-term synaptic facilitation. (**B**) Theta compression, that is correlation between peak correlation lag and distance of field centers in the control case. Each dot represents a field pair. Linear-circular regression line is indicated in black. Note that the sign of regression slope ($a$ in radians/cm) is determined by the directions of DG loop (negative in $\theta_{DG} = 180°$). (**C**) same as B, but for the lesion case. Theta compression is reduced as compared to the control condition. (**D**) Top: Distribution of spike phase during phase precession in all active (spike count > 5) cells in control and lesion case. Bottom: Distribution of phase precession range for all active cells.

cells receive more sensory depolarization, and thus, the movement-dependent extrinsic sequences are more activated. The model also explains, why pairs with similar preferred heading directions may be less extrinsic than pairs with approximately opposite preferred heading direction (dissimilar pairs), since the DG loop preferentially connects CA3 place cells with similar preferred heading directions. Both of the results follow the same trend as found in experimental data (*Yiu et al., 2022* report ratios of extrinsically to intrinsically correlated CA3 field pairs of 1.57 for both-best directions, 0.41 for both-worst directions, 0.87 for similar pairs and 2.43 for dissimilar pairs).

Thus, by introducing feedback excitation via the DG layer, intrinsic sequences are able to propagate in fixed directions on top of the movement-dependent extrinsic sequences. Theta sequence directionality is reflected through the change in spike correlation, which varies as a function of the difference between the direction of DG feedback and movement direction. The combination of extrinsic and intrinsic theta sequence activity is robust regarding changes of the parameters of short-term synaptic plasticity (*Figure 3—figure supplement 1*), as well as running speed (*Figure 3—figure supplement 2*), as long as place fields are wide enough to allow spikes in sufficiently many theta cycles.

## Lesion of DG reduces theta compression and phase precession range

One prediction of the DG-loop model, consistent with findings from DG lesion experiments (*Ahmadi et al., 2022*), is that DG would contribute to the temporal organization of spike sequences in CA3. To verify this hypothesis also in the model, we implemented a lesion of DG by disabling activity in the DG layer. To compensate for reduced excitatory drive caused by the lesion, we then increased probability of release of the sensory inputs thereby increasing the initial input amplitudes but removing short-term synaptic facilitation (*Figure 4A*).

We found that a DG lesion would reduce theta compression in sequence activity. Theta compression (*Dragoi and Buzsáki, 2006*) refers to the compression of seconds-long behavioural experience of place-field crossing into a neural representation of spike sequences at a shorter (theta) timescale. To quantify the strength of theta compression, we plotted the pair correlation lags versus the distance between the centers of two fields (abbreviated as 'lags' and 'pair distance' respectively), after simulating a rightward trajectory (*Figure 4B–C*). The magnitude of the linear-circular regression slope $a$ measures how much theta phase encodes a certain interval in space, and therefore, the strength of theta compression. As a result, theta compression is reduced for the DG-lesioned case ($a = 0.053$ radians/cm for both $\theta_{DG} = 0°$ and $\theta_{DG} = 180°$), as compared to the control case ($a = 0.183$ radians/cm for $\theta_{DG} = 0°$ and $a = -0.059$ radians/cm for $\theta_{DG} = 180°$) reproducing the finding (*Ahmadi et al., 2022*) that spatial encoding via theta sequences crucially depends on intact DG and suggesting that the loss of DG inputs could be compensated for by the increase of release probability in the spared afferent synapses from the MEC. The DG lesion also reduces the spike phase and phase range of phase precession (*Figure 4D*), which indicates the participation of DG loops in high-phase spiking. Both weaker phase precession and theta compression stress the important role of DG in temporal organization of CA3 sequences.

## Theta sequences in 2-d and out-of-field firing

So far, the model was only evaluated on bidirectional linear tracks, where running directions completely overlapped with the orientation of the DG loop connectivity. Now, we extend our analysis to 2-d space by examining oblique trajectories which cross the orientation of DG-loop projection at certain angles.

We first arrange the DG-loop connections such that the DG-loop orientation crosses a rightward trajectory at 45° and 225° (*Figure 5A–F*). Similar to the cases of $\theta_{DG} = 0°$ and $\theta_{DG} = 180°$ (*Figure 3D and H*), precession slopes are steeper and onsets higher when the trajectory direction aligns more with the orientation of the DG-loop, but with a smaller effect size for oblique crossings (*Figure 5A and C*) since DG-loop connectivity area only overlaps with part of the trajectory near the intersection. We further resolve the precession slope, onset and marginal phase for each place cell into 2-d maps (*Figure 5B and D*). Intrinsic sequences with a higher marginal spike phase can be clearly seen along the belt of DG-loop projections and are even extended to the outside of trajectory predicting 'off-track' spikes at high phases. Depending on the alignment between movement direction and DG-loop orientation, the slope becomes either more negative ($\theta_{DG} = 45°$) or more positive ($\theta_{DG} = 225°$). Analysis of extrinsicity and intrinsicity was conducted for all field pairs and confirmed the same trend as in *Figure 3* that best and dissimilar pairs are more extrinsic than worst and similar pairs, respectively

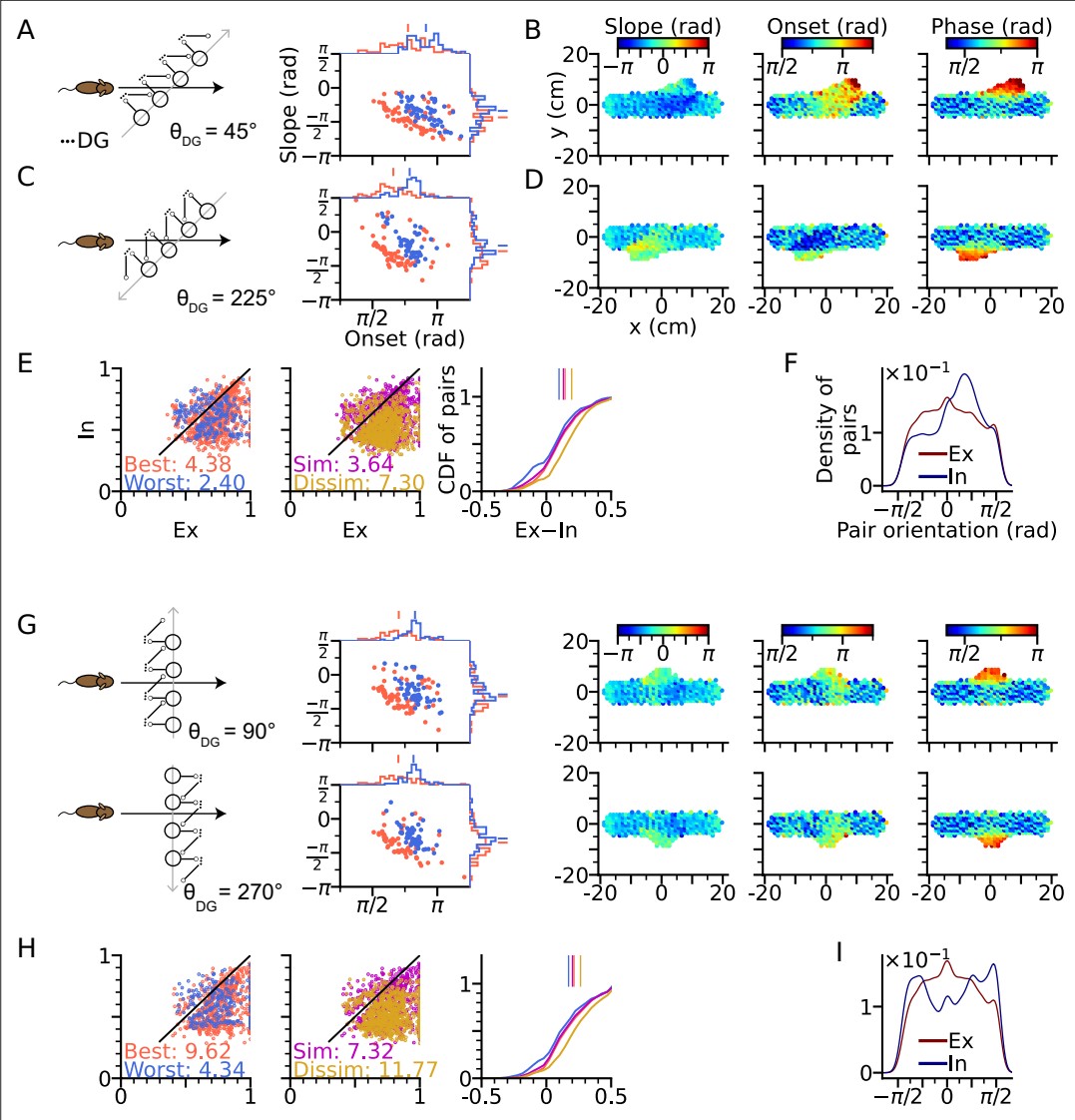

**Figure 5.** Intrinsic sequences lead to direction dependent 2-d phase precession and out-of-field firing (**A**) Left: Schematic illustration of DG-loop projection being tilted by 45° relative to the trajectory. Right: Distributions of phase precession onsets and slopes from the place cells along the trajectory as in *Figure 2G*. (**B**) Slopes (left), onsets (middle) and mean spike phases (right) of phase precession from the place cells as a function of field center. High spike phases and onsets occur along the DG-loop orientation where intrinsic spiking dominates and yield out-of-field firing (see the extrusions from horizontal dot clouds) with late onsets and phases. (**C–D**) Same as A-B, but DG-loop projection is at 225° relative to trajectory direction. (**D**) For DG loops pointing opposite to the sensorimotor drive, prospective firing along the DG loop yields less steep precession slopes and lower onset. (**E**) Extrinsicity and intrinsicity of all place field pairs along the trajectory as in *Figure 3J*. Some pairs are totally extrinsic (Ex = 1) because DG projection is absent at those parts of the trajectory. (**F**) Density of field pairs with extrinsic/intrinsic correlation as a function of the orientation of field center difference vector relative to the x axis. Intrinsic fields peak at 45°. (**G**) Same as A-D, but DG-loop orientations are perpendicular to the trajectory direction at 90° (top) and 270°. Prospective spikes from intrinsic sequences are initiated in the perpendicular directions. (**H**) Same as E, but with higher Ex-In ratios. (**I**) Field pairs with intrinsic correlations are at ±90°.

(*Figure 5E*). As a quantitative prediction, we computed the angle differences between field centers of cell pairs for the extrinsic and intrinsic populations, and observe that place field center differences in extrinsically correlated field pairs are mostly oriented horizontally (along the running direction) while place field center differences from intrinsically correlated field pairs are oriented along the DG-loop orientation $\theta_{DG} = 45°$, as by design (*Figure 5F*).

The analysis above is repeated for the geometric configurations that DG-loop connectivity is minimally interacting with the place cell activity induced by movement, that is when DG-loop orientation and the movement direction are perpendicular to each other ($\theta_{\text{DG}} = 90°$ and $\theta_{\text{DG}} = 270°$, *Figure 5G*). Similar effects as in *Figure 5B and D* on precession slope, onset and marginal phases are also observed in the 2-d map, except that the effects are further restricted to the intersection area in the middle. Also, the whole population has become more extrinsic as compared to the 45° and 225° cases (*Figure 5H*, see the numbers for extrinsic-intrinsic ratios) due to the smaller overlapping area between DG-loop projection and the trajectory. Lastly, the pair center difference orientation confirms that field pairs with extrinsic correlations follow the trajectory direction while those with intrinsic correlation are biased towards the DG-loop orientations (90°).

The results demonstrate the distinct roles of extrinsic and intrinsic sequences in 2-d spatial encoding. The former represents trajectory direction while the latter the associative memory towards specific locations. They can be played out at the same time separately in different directions and only interact with each other when they overlap. The interaction is reflected in directional dependence of phase precession properties, most notably the higher spike phases from the DG-CA3 recurrent input, as well as increased intrinsicity of pair correlation and extended firing fields along the orientation of the DG-loop projections. Intrinsic sequences also triggered out-of-field firing (*Figure 5B, D and G*) at late theta phases. In this case, the DG-loop connects to cells with remote place fields. These cells could even display multiple separated place fields, with high spike phases indicating the target location of the intrinsic sequence.

## Topology-free mechanisms of extrinsic phase precession

A well-known problem of phase precession models based on recurrent connectivity that applies to both, the original intrinsic *Tsodyks et al., 1996* and *Romani and Tsodyks, 2015* model, is that they do not explain how the topological connectivity matrix (in our case $W^{\text{CA3}}$) is generated (*Lisman and Redish, 2009*; *Jaramillo and Kempter, 2017*). Extrinsic theta sequences in a first exposure to a novel environment should therefore be missing. Although *Feng et al., 2015* find that theta sequences on a first exposure of a novel linear track are indeed much weaker (maybe only reflecting intrinsic sequences), their results nevertheless indicate a very fast learning time scale that is hard to reconcile with recurrent learning of a full spatial topology (particularly the generalization to 2-d). Also their result might be hampered by place field plasticity that biases the decoder towards backward-shifted place maps of later trials (*Parra-Barrero and Cheng, 2023*). We therefore explored, whether extrinsic 2-d sequences could also be generated by a model that is not relying on 2-d topology in the recurrent weights. To this end, we disabled the CA3 recurrence and compensated the missing level of excitation by an increased strength of the spatial input and the DG loops (see Methods section: Parameters of the models). Our simulations show that extrinsic sequences can still be generated by spatial input alone (*Figure 6A*), relying only on the short-term facilitation mechanism. Simulating CA3 activity with lesioned DG similarly abolishes the temporal organization of theta sequence and reduces the phase range (*Figure 6B*). The results demonstrate that the temporal order of extrinsic sequences could be coordinated solely by sensorimotor drive and does not necessarily require CA3 recurrence.

## Functional role of intrinsic sequences

While the function of extrinsic theta sequences in encoding the actual trajectory of an animal (connecting the recent past, present and near future locations) is obvious, the potential role of the less readily apparent intrinsic sequences is not straight forward. Simulations of trajectories in 2-d (*Figure 5*) suggest intrinsic activity may serve a role to identify certain location-direction pairs independent of the current trajectory. Here, we follow this idea by evaluating the hypothesis that the intrinsic sequences signal a stable 'landmark' (location/direction pair) cue by a temporal code that is invariant to different directions of approach.

To test our hypothesis, we constructed a downstream readout neuron that would reliably identify the presence of the intrinsic sequence independently of the animal's running direction, whereas it would not be able to do so for only extrinsic sequences. To this end, we trained the synaptic weights using the tempotron learning rule (*Gütig and Sompolinsky, 2006*), which is able to implement binary classification based on temporal relations of input spike patterns (see Methods section: Tempotron). Two tempotrons were trained to recognize the spike patterns from the place cells, one

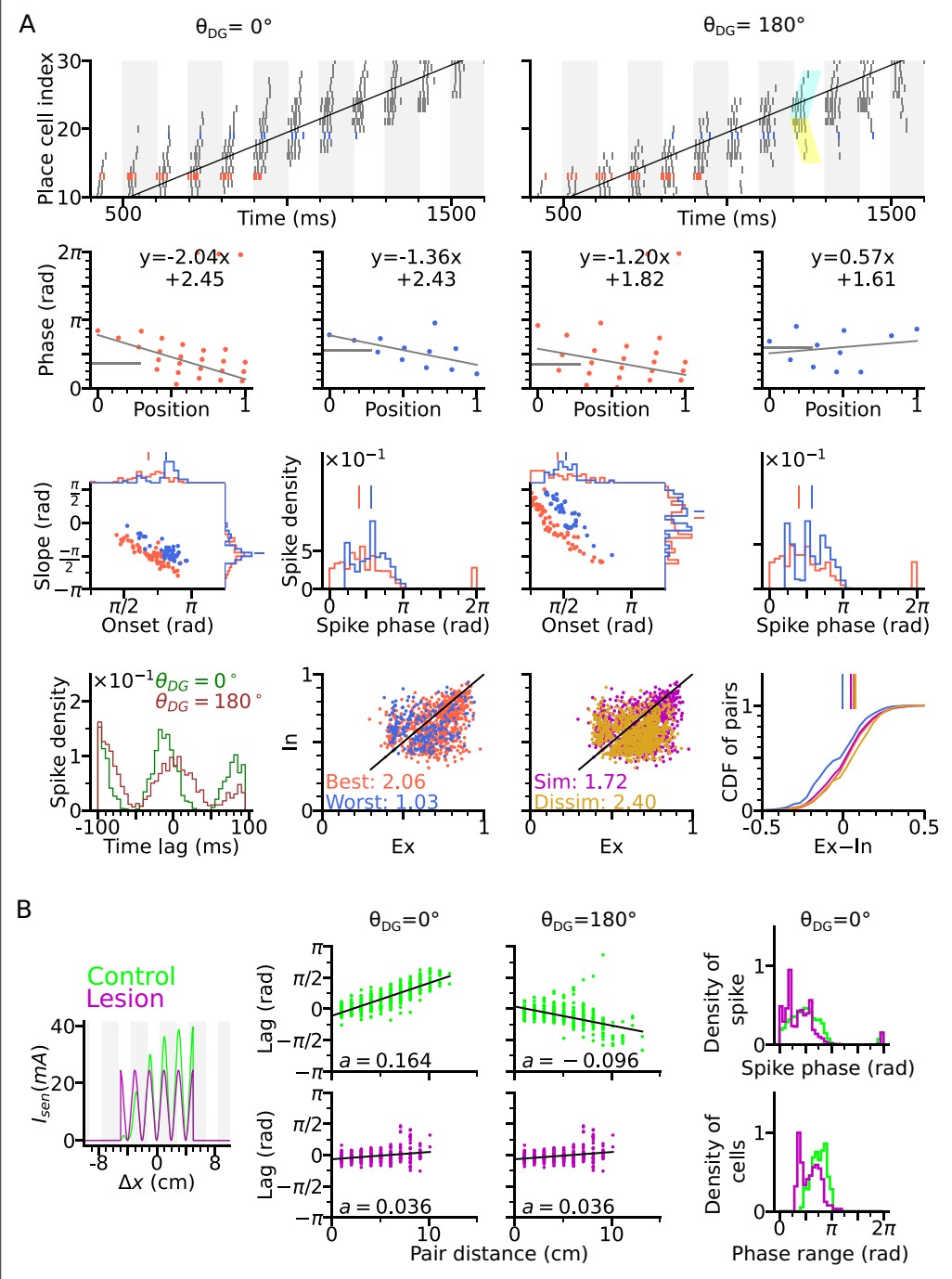

**Figure 6.** Extrinsically driven theta correlations can be temporally organized by sensorimotor drive alone without CA3-CA3 recurrence. Simulations were performed without CA3-CA3 recurrence but with stronger spatial input. (**A**) Same as *Figure 3*. Extrinsically driven theta correlations and phase precession are still present. (**B**) Same as *Figure 4*. DG is still integral to the theta compression in a network model without CA3-CA3 recurrence.

taking input from a model with DG-loop connectivity at $\theta_{DG} = 0°$, and one without DG-loop connectivity to serve as a control only having access to extrinsic sequences (*Figure 7A*). Non-moving spatial inputs were applied to the CA3 place cells at the centers of with-loop and no-loop populations and their spike patterns in subsequent theta cycle were used as training patterns, mimicking a situation in which network activity is evoked without sensory-motor input as, for example in a offline situation before the animal walks or maybe even has seen the environment. The training patterns have only

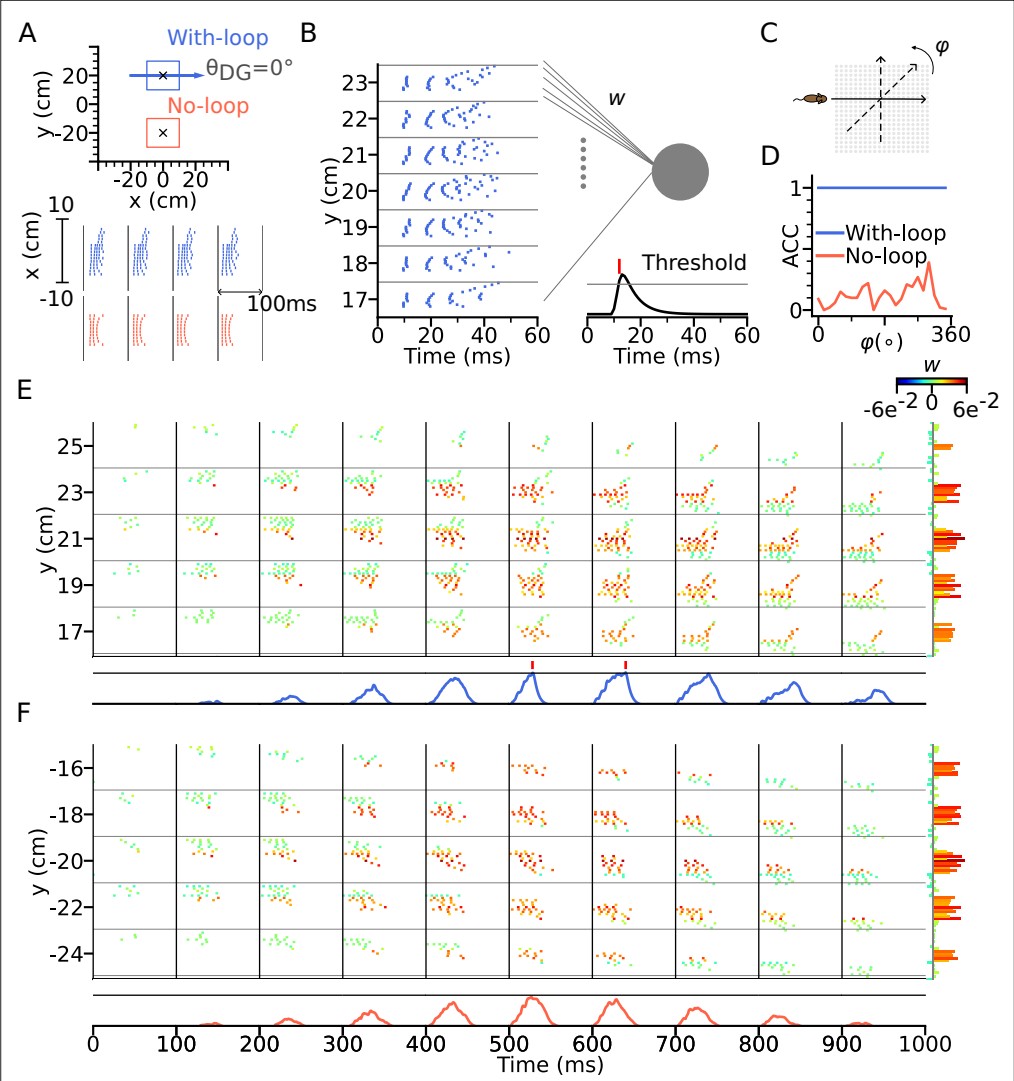

**Figure 7.** Intrinsic sequences provide a stable landmark for positional decoding using a tempotron. (**A**) Top: Two tempotrons are trained for place cell populations within the top (with DG loop; blue) and bottom (no DG loop; red) squares, to recognize the presence of the corresponding sequence activities. DG-loop rightward projection is indicated by blue arrow and only exists in the blue square. Non-moving spatial inputs are applied to the CA3 network centered at the two locations (marked by black crosses) to evoke spike sequences for training. Bottom: Resulting spikes of the place cell network zoomed in to the subset of field centers from x=-10 to x=10 for y=+20 (with-loop, top raster plot) and y=-20 (no-loop, bottom). Each theta cycle is one (+) training pattern, which the tempotron is trained to detect by eliciting a spike. (**B**) Example training pattern with spikes of place cells from x=-10 to x=10 (in each rectangular row) fixed at different values of y. Only one theta cycle is shown. Each place cell delivers spikes to the dendrite of the tempotron, producing post-synaptic potentials (PSPs) at the soma (line plot at the bottom). Synaptic weights are adapted by the tempotron learning rule such that PSPs can cross the threshold (gray line) and fire for the detection of the sequence. After the tempotron has fired, the PSPs will be shunted. (**C**) Sequence detection is tested while the simulated animal ran on a trajectory with varying direction ($\varphi$) from 0° to 360° with a 15° increment to detect the presence of the sequence. (**D**) Detection accuracies (ACC) for with-loop (red line) and no-loop (blue) input populations. Note that the tempotron cannot detect the no-loop sequences when tested on trajectories at various angles. (**E**) Detection of the intrinsic sequence for a trajectory $\varphi = 180°$ for the DG-loop condition. Spike raster is shown for every two horizontal rows of place cells in the arena and color-coded by the synaptic weights (see color bar on the right). Tempotron soma potential is shown at the bottom for each pattern. (**F**) Same as E, but for no-loop inputs. The tempotron remains silent.

The online version of this article includes the following figure supplement(s) for figure 7:

**Figure supplement 1.** Decoding of positional landmarks using tempotrons in a network model without CA3-CA3 recurrence.

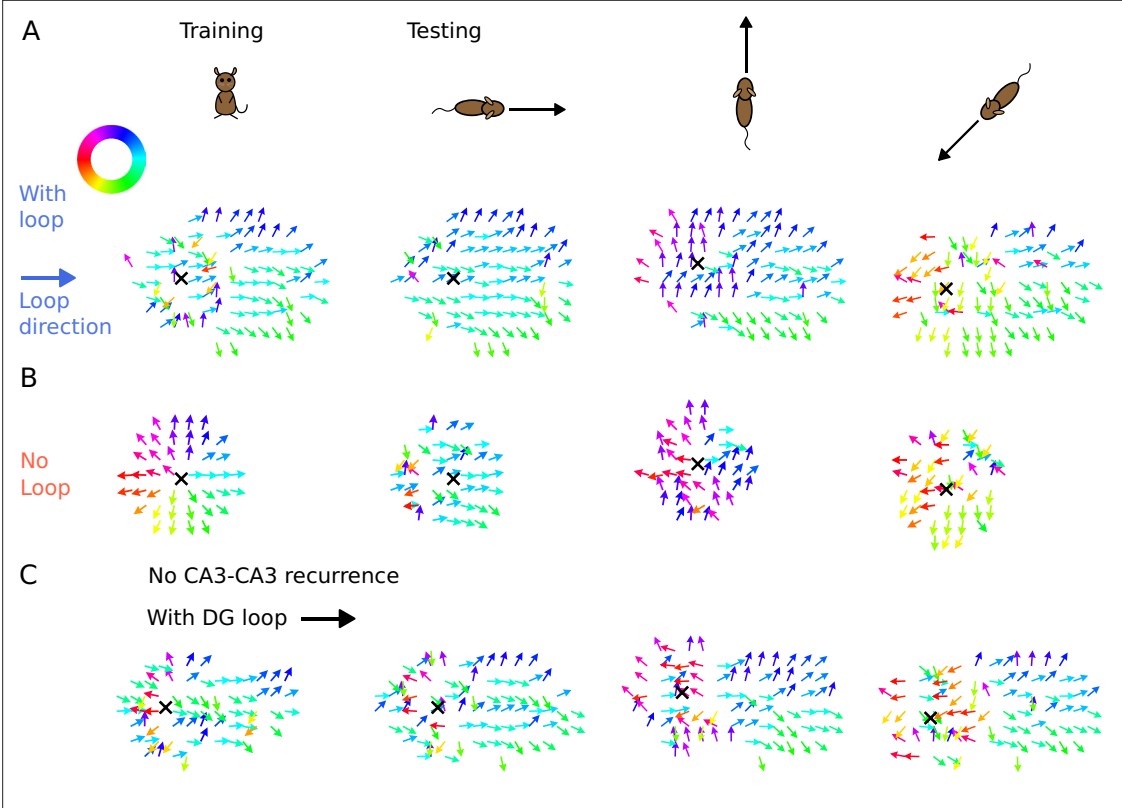

**Figure 8.** Illustration of spike time gradients in one theta cycle (500–600 ms) with and without DG loop. (**A**) Time gradients with a DG loop projecting to the rightward direction ($\theta_{DG} = 0°$). Each arrow is located at a CA3 place field center. The arrow direction indicates the spike time gradient, equivalently the 'travelling direction' of sequence activity, which is calculated as the sum of the directions to the 8 neighbouring field centers, weighted by the difference between their mean spike times in one theta cycle. Arrow direction is color-coded according to the color wheel. Black cross marks the instantaneous position of the animal. The first column shows the training condition when a non-moving spatial stimulus is applied. The three columns on the right show the testing condition when the rat is running in various directions. The sequence mostly propagates rightwards, following the DG-loop direction even when the animal runs in different directions. (**B**) Same as A without a DG loop. Sequences propagate outward from the animal position as a concentric travelling wave during training. During testing, spike time gradients follow the running direction. (**C**) Same as A, using the network model without CA3 recurrence. As in *Figure 6*, the extrinsic sequence is driven solely by the STF mechanism of the spatial input. Intrinsic sequences in this model still remain invariant to running directions and function as spatial landmarks.

(+) labels, which the tempotrons are trained to recognize by firing a spike (*Figure 7B*). We then test the tempotrons with spike patterns induced by the animal running on different trajectories through the trained location with running directions varying between 0° to 360° (*Figure 7C*). All spikes in each training and testing pattern are individually jittered by adding a noise term $\sigma \sim \mathcal{N}(0, (2\,\mathrm{ms})^2)$ for 100 times, producing 100 samples for each pattern. The tempotron is said to successfully recognize the sequence of a trajectory direction if any of the theta cycles throughout the trajectory elicits a spike, that is while running, the readout cell would evaluate the place cell sequence in every theta cycle for information on the trained landmark.

We found that the tempotron trained on the intrinsic sequence from the DG loop is able to recognize the sequence patterns produced for all running directions, while the tempotron trained without a DG-loop fails to identify the extrinsic sequences most of the time (see accuracies in *Figure 7D*). The reason is that the spike patterns induced by intrinsic sequences remain similar to the training pattern despite being approached in other directions (see sequential contributions in *Figure 7E*), while spike patterns for the no-loop condition are different between training and testing (*Figure 7F*). The distinction is further illustrated in *Figure 8*, where 2-d maps of spike time gradients in one theta cycle are plotted with respect to running direction/training condition. Sequences always contain components that propagate along the projection direction of DG loop, while, without such a loop, they only propagate along the running direction. Moreover, during training, the no-loop condition evokes concentric

waves, reflecting the 2-d topology of the recurrent weights. Similar results were also achieved without CA3-CA3 recurrence (*Figure 7—figure supplement 1* and *Figure 8C*).

Our results show that intrinsic sequences can provide a stable correlation signal which allows reliable decoding of locations through temporal correlations. The intrinsic temporal code remains detectable even when mixed with extrinsic sequences.

## Discussion

We presented a model of hippocampal theta sequences in 2-d environments, suggesting that both extrinsic and intrinsic mechanisms are required to explain experimental reports that phase precession and spike timing correlations are non-homogeneous across running directions. Although phase precession already becomes directional by including direction-dependent sensory input into a purely extrinsic model, directionality of spike timing correlations cannot be explained by such a model. We, however, demonstrated that the correlation preference could be implemented by fixed recurrent loops via a model DG layer. We further supported the model assumptions by showing that DG lesions plus compensatory sensory drive can abolish the theta compression effect in CA3 spiking activity (*Ahmadi et al., 2022*). By employing a spike-based temporal pattern decoder (tempotron), we showed that the intrinsic sequences could function as stable signatures that act as anchors of the spatial code.

Early intrinsic models (*Tsodyks et al., 1996*) were challenged owing to their inability to generate phase precession in backward travel (*Figure 1E*, also see *Cei et al., 2014*), as well as the predominantly extrinsic correlations observed in CA1 (*Huxter et al., 2008*). In our hybrid model, phase precession still occurs during backward travel ($\theta_{DG} = 180°$) but at a lower probability as indicated by the larger fraction of positive phase-position slopes (*Figure 3H*). Also, extrinsic sequences still dominate over intrinsic sequences as indicated by the majority of field pairs being extrinsic (*Figure 3J*). Both the reduced expression of phase precession in reverse runs and the dominance of extrinsic sequences are in accordance with the experimental data (*Yiu et al., 2022*).

The mixture of extrinsic and intrinsic mechanisms in our theory, naturally gives rise to the directionality of spike correlations and phase distributions. As the trajectory aligns itself with the DG loops, the ratio of intrinsic to extrinsic sequences increases. As a result, spike correlations become more rigid and the phase distribution is shifted upward due to the accumulated synaptic transmission delay from the reverberating activity between CA3 and DG populations. Adding directional sensory input activates extrinsic sequences in the best direction more strongly, and hence, leads to an association between best-angle (worst-angle) pairs and extrinsicity (intrinsicity). These predictions of our model are corroborated by past reports of higher spike phases in the non-preferred arm of a T-maze (*Kay et al., 2020*) as well as the association of rigid correlations with upward shifts in spike phases and an increase in worst-angle pairs (*Yiu et al., 2022*). The experimental distinction between extrinsic and intrinsic components in theta sequences has so far only been achieved in pairs of place cells, owing to the limited number of simultaneously recorded place cells with overlapping fields. Our model predicts that similar distinctions should also be observable in higher-order statistics, obtained from overlapping fields of a larger number of cells. Instead of correlation lags, we suggest to use temporal pattern detection methods (e.g. *Chenani et al., 2019*) to unveil the respective sequence contributions.

Since intrinsic sequences can also propagate outside the trajectory (out-of-field firing in *Figure 5*) and activate place cells non-locally, our model predicts direction-dependent expansion of place fields, or even multiple place fields, with the intrinsic sequence's target location exhibiting late spike phases and higher phase precession onsets. Remote activation during locomotion has already been observed in a previous study (*Sasaki et al., 2018*) where CA3 place cells preferentially firing at one arm of the maze were also activated at reward locations at other arms. In our model, only short-range intrinsic connectivity was considered, thus, place field boundaries expand locally but in a skewed manner matching the sequence direction. Skewness of place fields has been reported by a number of studies (*Mehta et al., 1997*; *Shen et al., 1997*; *Mehta et al., 2000*; *Ekstrom et al., 2001*; *Lee et al., 2004*; *Burke et al., 2008*; *Cei et al., 2014*; *Roth et al., 2012*; *Dong et al., 2021*) showing place fields to be asymmetrically expanded opposite to the direction of travel. This effect was connected to plasticity as it develops after repeated traversal, and due to its dependence on NMDA receptor activation (*Ekstrom et al., 2001*; *Burke et al., 2008*; *Shen et al., 1997*). These plasticity studies show that the hippocampal place code is shaped by intrinsic synaptic computations including temporal activation patterns in theta sequences (*Feng et al., 2015*). Apart from being conducted on linear tracks and not

2-d environments, most of this work focused on CA1 and associated Schaffer collateral plasticity. Yet some prior studies (*Lee et al., 2004*; *Roth et al., 2012*) did show that place fields in CA3 were more skewed than in CA1, which our model would explain by CA3 expressing more intrinsic sequences than CA1 consistent with prior experimental observations (*Yiu et al., 2022* reported ratios of extrinsically to intrinsically driven cell pairs of 1.44 in CA1 and 1.23 in CA3).

A further prediction of hard-wired DG loops is that the resulting activity patterns (intrinsic sequences) should not remap under conditions of global or partial remapping (*Leutgeb et al., 2004*). Instead the same intrinsic sequence components should be observable in multiple environments, however, they might only be seen in a small fraction and thus this prediction is potentially hard to test.

The back-projection from CA3 to DG is a crucial anatomical prerequisite of our model, but was rarely explored compared to the feed-forward inputs via the perforant pathway. The proposed CA3-DG recurrent structure of this model, albeit simplified, is consistent with the anatomical evidence. Pyramidal cells in CA3 innervate the mossy cells at the DG hilus (*Scharfman, 1994*; *Scharfman, 2016*), which then project to granule cells through both excitatory and inhibitory pathways (*Hsu et al., 2016*; *Scharfman, 1995*; *Larimer and Strowbridge, 2008*; *Soriano and Frotscher, 1994*), and subsequently back to CA3 pyramidal cells. An optogenetic study (*Hsu et al., 2016*) showed that the net effect of mossy cells on granule cells was predominantly inhibitory, suggesting that the DG ensembles excited by mossy cell synaptic drive are sparsified by suppressing unwanted out-of-ensemble activity. Indeed, past studies showed that reliable excitatory effect could be observed when granule cells were depolarized (*Scharfman, 1995*) and when they received back-propagation of sharp wave bursts from CA3 population (*Penttonen et al., 1998*). This indicates that the excitatory recurrent pathway from CA3 via DG exists and might allow activity reverberation between two layers. While our model, owing to its simplicity and generality does not require any DG specific pathways and would work equally well with any other anatomical interpretation of the CA3 feedback, we hypothesize the intrinsic feedback connectivity to arise via the DG, particularly because DG lesions were shown to eliminate the coordinated temporal structure of CA3 activity and to be instrumental to sequence organization (*Figure 4* and *Ahmadi et al., 2022*).

Our model assumed a connectivity pattern in the DG loops, in which neurons activate the neighbours along a specific direction, as inspired by Hebb's phase sequences (*Hebb, 1949*) and, hence, replay of the loop would activate a spatially plausible virtual trajectory. The loop connectivity could either arise from previous learning, or might be present already beforehand (*Dragoi and Tonegawa, 2013*), with spatial topology inherited by associating 2-d sensory features to cell ensembles in the loop (*Leibold, 2020*). The resulting topology can exhibit discontinuous long-range jumps to other locations (*Sasaki et al., 2018*) or consist of a discrete set of (behaviorally relevant) locations (*Pfeiffer, 2022*).

Different from other phase precession models, we also included heading direction as part of the sensory input, as inspired by past literature that CA1 (*Markus et al., 1995*; *Acharya et al., 2016*; *Stefanini et al., 2020*), CA3 (*Mankin et al., 2019*), and DG place cells (*Stefanini et al., 2020*) exhibit directional selectivity in firing rates, potentially inherited from the upstream head-direction cells in the medial entorhinal cortex (*Giocomo et al., 2014*) and postsubiculum (*Taube et al., 1990*). As a result, the directional drive immediately translates to phase directionality in theta sequences, partly contributing to the upward shift of the phase distribution in the worst angles. Such phase directionality arises naturally from the intracellular dynamics of a spike-based model, where stronger depolarization causes earlier spiking. This phase-rate dependence has already been used in previous models (*Harris et al., 2002*; *Mehta et al., 2002*; *Thurley et al., 2008*), where the increasing depolarization within place fields directly relates to decreasing spike phases. The causal effect of firing rate on spike phases, however, was disputed by *Huxter et al., 2003* as they showed that precession slopes and spike phases remained the same between high- and low-spiking runs, suggesting that the phase is not single-handedly determined by firing rate. In our model, firing rate is determined by both low-phase spiking from sensory input and high-phase spike arrivals of DG-CA3 loops, both producing opposing effects on the phase distribution. Thus, depending on the strength and geometry of the DG-CA3 connectivity, spike phases are not fully determined by firing rate.

By using a tempotron to decode the spike patterns, we show that the spike patterns of intrinsic sequences can serve as a stable landmark which remains decodable across multiple running directions. The invariant temporal patterns could serve as anchors of spatial memories in a novel environment,

since place fields only stabilize after the animal becomes familiar with the environment (*Wilson and McNaughton, 1993*). The pre-existing sequence motifs, even at times when the spikes of the neurons are not spatially tuned to a location, can still encode the position based on their temporal relations alone. The idea has previously been spelled out (*Cheng, 2013*) and numerically verified (*Leibold, 2020*; *Parra-Barrero and Cheng, 2023*) with multiple fixed sequences that form a decodable spatial representation.

Intrinsic sequences may thus act as a scaffold around which a new spatial code can be built for new but similar behavioral contexts, where similarity could for example be identified by a salient feature. Once the behavioral context of a situation changes, new intrinsic sequences would be observable. These intrinsic landmarks need to be stable across time, as shown for some dentate gyrus representations (*Hainmueller and Bartos, 2018*). We speculate that offline sequences observed during replay and preplay (for review see *Buhry et al., 2011*; *Dragoi and Tonegawa, 2014*), would correspond to the intrinsic activity patterns and indicate the context expectation of an animal (which can be detected by a tempotron). The functional roles of intrinsic sequences may thus not be limited to spatial memories. While, in the spatial domain, intrinsic sequences could be interpreted as spatial trajectories (*Kay et al., 2020*; *Sasaki et al., 2018*), virtual non-spatial trajectories could represent working memories contents (*Jensen et al., 1996*) available for general decision making processes.

# Methods

## Key resources table

| Reagent type (species) or resource | Designation | Source or reference | Identifiers | Additional information |
|---|---|---|---|---|
| Software, algorithm | Python | Python Software Foundation | https://www.python.org/ RRID:SCR_008394 | |
| Software, algorithm | Linear-circular regression | *Kempter et al., 2012* | | The algorithm is customized to our analyses |
| Software, algorithm | Tempotron | *Gütig and Sompolinsky, 2006* | | The algorithm is customized to our analyses |

## Neuronal model

Generation of neuronal action potentials is modelled according to *Izhikevich, 2003*. The soma potential $v$ and the adaptation variable $u$ of unit $i$ at time $t$ (in ms) follows the equations:

$$\dot{v}_i(t) = 0.04v_i^2(t) + 5v_i(t) + 140 - u_i(t) + I_i(t)$$

$$\dot{u}_i(t) = a\left[b\,v_i(t) - u_i(t)\right]$$

$$I_i(t) = I_i^R(t) + I_i^S(t) - I^\theta(t)$$

Any time $v(t)$ crosses the threshold 30 mV from below, we register a spike for the neuron and reset the soma potential by $v(t) \leftarrow c$ and the adaptation variable by $u(t) \leftarrow u(t) + d$. For the excitatory pyramidal place cells, we use parameters $a = 0.035$, $b = 0.2$, $c = -60$ mV, $d = 8$, which provides the neuron with burst firing characteristics. For the inhibitory interneurons, the parameters were $a = 0.02$, $b = 0.25$, $c = -65$ mV, and $d = 2$, which corresponds to fast spiking patterns. $I(t)$ is the total sum of recurrent $I^R(t)$, sensory $I^S(t)$ and oscillatory theta input

$$I^\theta(t) = 7\left[1 + \cos\left(\frac{2\pi t}{100\text{ ms}}\right)\right]/2$$

We chose to use the phenomenological spike generation model of *Izhikevich, 2003*, since it allows to adjust burst firing properties with only few parameters that efficiently emulate the bifurcation structure of spike generation. Synaptic integration below threshold is not affected by the spike generation model and will thus be treated by conventional synaptic models.

## Spatial input

The place field centers $\mathbf{p}_i^{CA3} = \left[x_i^{CA3}(t), y_i^{CA3}(t)\right]$ of $80 \times 80 = 6400$ excitatory CA3 cells equally tile the 80 by 80 cm square arena. Place cell firing rates are modelled direction-sensitive, with preferred heading

directions $\psi_i^{CA3}$ semi-randomized among each 2×2 tile of place cells by randomly rotating a set of four equally spaced direction angles by a uniformly distributed angle $\xi$, that is

$$\left[\psi_i^{CA3}, \psi_{i+1}^{CA3}, \psi_{i+2}^{CA3}, \psi_{i+3}^{CA3}\right] = \left[0°, 90°, 180°, 270°\right] + \xi \mod 360°.$$

The sensory input $J_i^S(t)$ into the $i$-th neuron depends on the instantaneous position, $\mathbf{p}(t) = \left[x(t), y(t)\right]$, and heading direction $\psi(t)$ of the animal as

$$J_i^S(t) = \begin{cases} A_i^S(t)\, I^{MEC}(t) & \text{if } d\left(\mathbf{p}(t), \mathbf{p}_i^{CA3}\right) \leqq 5 \text{ cm} \\ 0 & \text{if } d\left(\mathbf{p}(t), \mathbf{p}_i^{CA3}\right) > 5 \text{ cm} \end{cases}$$

$$A_i^S(t) = A^{pos} + A^{dir} \exp\left(\cos\left(\psi(t) - \psi_i^{CA3}\right) - 1\right)$$

$$I^{MEC}(t) = \frac{1}{2}\left[1 + \cos\left(\frac{2\pi t}{100 \text{ ms}} + 70° \frac{\pi}{180°}\right)\right],$$

where $A^{pos}$ is the amplitude of positional tuning and $d(\cdot)$ computes the Euclidean distance between two positions. The positional tuning curve is implemented as a rectangular box function, where the place cell only receives sensory input if the animal is within 5 cm from the field center. Directional tuning is implemented as an additional amplitude gain $A^{dir}$ to the positional current depending on the circular difference between the animal's heading and the neuron's preferred heading direction $\psi_i^{CA3}$. The sensory input is assumed to be modulated by theta oscillations from medial entorhinal cortex (MEC) $I^{MEC}(t)$ with a phase shift of 70° (**Mizuseki et al., 2009**).

The sensory input $J^S$ is subsequently transformed to the input current $I^S$ via short-term facilitation (STF)

$$\dot{s}_i^F(t) = \frac{(S_0^F - s_i^F(t))}{\tau^F} + (S_1^F - s_i^F(t))\Phi^F J_i^S(t)$$

$$I_i^S = J_i^S(t)\left[s_i^F(t)\right]^2,$$

where the facilitation variable $s_i^F$ decays to $S_0^F$ with a time constant $\tau^F = 500$ ms and increases to $S_1^F$ when the sensory input $J_i^S$ is present. The time constant $\tau^F$ of facilitation of neocortical synapses was in the range suggested by **Tsodyks et al., 1998** following previous experimental reports (**Mejías and Torres, 2008**; **Zucker and Regehr, 2002**). $\Phi^F$ controls the strength of the STF. The facilitation variable is squared to include non-linear interactions in presynaptic calcium dynamics. As a result, facilitated sensory input $I_i^S$ increases over time and becomes stronger in the later part of the field, thus effectively generating a spatially graded input strength.

Note that only the CA3 place cells receive the sensory input. $I_i^S(t)$ is not applied to the place cells in DG and all of the inhibitory interneurons.

## CA3 recurrent connections

Place cells in CA3 connect with each other by excitatory synapses. The excitatory synaptic current $I_i^E(t)$ is conductance-based, and follows the equations:

$$\dot{g}_i^E(t) = \frac{-g_i^E(t)}{\tau^E} + \frac{1}{N_J}\sum_{j,f} W_{ij}\, s_j^D(t)\, \delta(t - t_j^{(f)} - \tau_0) \tag{1}$$

$$I_i^E(t) = \left[V^E - v_i(t)\right]g_i^E(t) \tag{2}$$

The conductance $g_i^E$ of a post-synaptic cell $i$ is increased by the spike arrivals at times $t_j^{(f)}$ from the pre-synaptic cell $j$, and decay with a time constant $\tau^E = 12$ ms. $N_J = 6,400$ is the number of presynaptic place cells, $V^E = 0$ mV is the reversal potential of the excitatory synapses and $\tau_0 = 2$ ms is the synaptic transmission delay.

The synaptic weights $W_{ij}$ from cell $j$ to cell $i$ depend on the distance between place cell centers and on the similarity of their preferred heading angles, i.e.,

$$W_{ij}^{\text{CA3}} = J_{ij} \left\{ B^{\text{pos}} + B^{\text{dir}} \exp\left[ K^{\text{CA3}}\left( \cos(\psi_i^L - \psi_j^L) - 1 \right) \right] \right\} \exp\left( \frac{-d(\mathbf{p}_i^{CA3}, \mathbf{p}_j^{CA3})^2}{2\sigma^2} \right) ,$$

where $B^{\text{pos}}$ and $\sigma = 2$ cm correspond to the maximum strength and width of the location-specific interaction, respectively. $B^{\text{dir}}$ and $K^{\text{CA3}}$ control the maximum strength and the concentration of the directional dependence, respectively. $J_{ij}$ models the rightward asymmetry of the cell connections, which was only turned on when we simulated the 2-d variant of *Tsodyks et al., 1996* model in *Figure 1C–F* and otherwise turned off in the rest of our analysis.

If rightward asymmetry is ON, $J_{ij} = 1$ if $x_j^{CA3} < x_i^{CA3}$, else $0$

If rightward asymmetry is OFF, $J_{ij} = 1$

Furthermore, the recurrent synaptic conductances underwent short-term synaptic depression (STD), as was proposed in *Romani and Tsodyks, 2015* to serve as sequence generator in 2-d space. The mechanism penalizes the recurrent input into the place cells behind the animal. As a result, the differential recurrence strengths translate to a gradient of spike phases and produces extrinsic sequences in the direction of travel. We model the STD by the variable $s_i^D(t)$ which represents the available synaptic resource and follows the dynamics:

$$\dot{s}_i^D(t) = \frac{1 - s_i^D(t)}{\tau^D} - U^D \delta(t - t_i^{(f)}) ,$$

where $s_i^D$ recovers to 1 with a time constant $\tau^D = 500$ ms and is depleted by a fraction $U^D$ every time a spike occurs. The recovery time constant is comparable to experimentally obtained values of cortical neurons (200–800 ms in *Tsodyks and Markram, 1997*; *Markram et al., 1998*; *Abbott et al., 1997*; *Zucker and Regehr, 2002*) and previous modelling work (450–800 ms in *Romani and Tsodyks, 2015*; *Haga and Fukai, 2018*; *Tsodyks and Markram, 1997*; *Tsodyks et al., 1998*). The STD only applies to synaptic connections when presynaptic cells are CA3 place cells. $s_i^D(t)$ is fixed at 1 when the presynaptic cells are inhibitory interneurons or DG place cells.

## DG layer

We simulated $N_{\text{DG}} = 40 \times 40 = 1600$ place cells in the DG layer, with place field centers equally tiling the environment. The DG cells do not receive sensory input. Their positional ($x_i^{\text{DG}}, y_i^{\text{DG}}$) and directional ($\psi_i^{\text{DG}}$) tunings are determining synaptic strengths to and from the CA3 layer. The directional tuning is semi-randomized as described for CA3. The synaptic current dynamics follow *Equations (1) and (2)*. Excitatory synaptic weights from CA3 place cells to DG place cells are defined as

$$W_{ij}^{\text{CA3−DG}} = C_j^{CA3} B^{\text{DG}} \exp\left[ K^{\text{DG}}\left( \cos(\psi_i^{\text{DG}} - \psi_j^{\text{CA3}}) - 1 \right) \right] \exp\left( \frac{-d(\mathbf{p}_i^{DG}, \mathbf{p}_j^{CA3})^2}{2\sigma^2} \right) ,$$

which are dependent on the differences in the place field centers and preferred heading angles between the CA3 and DG populations. The variable $C_j^{\text{CA3}}$ strengthens outgoing connections from CA3 place cells on the path corresponding to the intrinsic sequence by choosing

$$C_j^{\text{CA3}} = \max_{k \in [-10,10]} \left\{ \exp\left( \frac{-d(\mathbf{p}_k^C, \mathbf{p}_j^{CA3})^2}{2\sigma^2} \right) \right\} ,$$

where $\mathbf{p}_k^C$ varies with the intrinsic path direction $\theta_{DG}$ as $\mathbf{p}_k^C = [2k \cos(\theta_{\text{DG}}), 2k \sin(\theta_{\text{DG}})]$.

The excitatory synaptic strengths from DG to CA3 are chosen such that DG cells project back to CA3 cells with place field centers shifted by a vector $\mathbf{r} = [4\cos(\theta_{\text{DG}}), 4\sin(\theta_{\text{DG}})]$ of fixed length of 4 cm along the intrinsic path, that is

$$W_{ij}^{\text{DG−CA3}} = B^{\text{DG}} \exp\left[ K^{\text{DG}}\left( \cos(\psi_i^{\text{CA3}} - \psi_j^{\text{DG}}) - 1 \right) \right] \exp\left( \frac{-d(\mathbf{p}_i^{CA3} - \mathbf{r}, \mathbf{p}_j^{DG})^2}{2\sigma^2} \right) . \tag{3}$$

The model has no synaptic connections between DG excitatory neurons.

## Inhibitory synapses

The model additionally contains $N_I = 250$ inhibitory interneurons (denoted as Inh) each for the CA3 and the DG layer. They provide inhibitory feedback separately to the excitatory cells within each layer (CA3-Inh-CA3 and DG-Inh-DG). The dynamics of their synaptic currents mirrors the excitatory synapses, that is

$$\dot{g}_i^I(t) = \frac{-g_i^I(t)}{\tau^I} + \frac{1}{N_I} \sum_j^{N_I} W_{ij}^{X-Y} \delta(t - t_j^{(f)} - \tau_0)$$

$$I_i^I(t) = \left[ V^I - v_i(t) \right] g_i^I(t),$$

with $\tau^I = 10\,\text{ms}$, $V^I = -80\,\text{mV}$. CA3 and DG have all-to-all connections to their inhibitory populations with uniformly randomized strengths, i.e. $W_{ij}^{X-Y} = W_0^{X-Y}\xi$, with $\xi \sim \mathcal{U}(0,1)$ is the maximum synaptic strength, and the notation X-Y corresponds to Inh-CA3 and Inh-DG connections. There is no synaptic connection between inter-neurons, that is $W^{\text{Inh}-\text{Inh}} = 0$.

The total recurrent current entering each excitatory neuron is thus the sum of the excitatory and inhibitory current:

$$I_i^R(t) = I_i^I(t) + I_i^E(t)$$

**Table 1.** Model parameters used in simulations according to Figure panels.
In, Ex, C. and L. refer to intrinsic, extrinsic, control and lesion respectively.

| Name \ Figure | 1 (In) | 1 (Ex) | 2 | 3 | 4 (C.) | 4 (L.) | 5 | 6A | 6B (C.) | 6B (L.) | 7 |
|---|---|---|---|---|---|---|---|---|---|---|---|
| $A^{\text{pos}}$ | 7.5 | 9.0 | 6.5 | | | | | 9.5 | 7.5 | | 6.5 |
| $A^{\text{dir}}$ | 0 | 6 | | | | | | 9 | 8 | | 6 |
| $S_0^F$ | 1 | | 0 | | 1.25 | 0 | | 0.25 | 0 | 1.25 | 0 |
| $S_1^F$ | 1 | | 2 | | 1.25 | 2 | | 1.5 | 2 | 1.25 | 2 |
| $\Phi^F$ | 0 | | 0.001 | | 0 | 0.001 | | | | 0 | 0.001 |
| $B^{\text{pos}}$ | 1100 | 0 | | | | | | | | | |
| $B^{\text{dir}}$ | 0 | | 2000 | 1500 | | | 2000 | 0 | | | 1500 |
| $K^{\text{CA3}}$ | 0 | | 1 | | | | | | | | |
| $J_{ij}$ | ON | OFF | | | | | | | | | |
| $U^D$ | 0 | 0.9 | 0.7 | | | | | | | | |
| $N_{\text{DG}}$ | 0 | | 40×40 = 1600 | | | | | | | | |
| $B^{\text{DG}}$ | 0 | | | 3000 | 0 | 4000 | | | 4000 | 0 | 4000 |
| $K^{\text{DG}}$ | 0 | | | 1 | | | | | | | |
| $N_I$ | 0 | | 250 | | | | | | | | |
| $W_0^{\text{CA3}-\text{Inh}}$ | 0 | | 50 | | | | | | | | |
| $W_0^{\text{Inh}-\text{CA3}}$ | 0 | | 5 | | | | | | | | |
| $W_0^{\text{DG}-\text{Inh}}$ | 0 | | | 350 | | | | | | | |
| $W_0^{\text{Inh}-\text{DG}}$ | 0 | | | 35 | | | | | | | |

### Excitatory synapses to interneurons

Interneurons only receive all-to-all excitatory currents from their respective layer. Those currents are modelled according to *Equations 1; 2*. The synaptic weights are constant and denoted by $W_0^{\mathrm{CA3-Inh}}$ and $W_0^{\mathrm{DG-Inh}}$.

### Parameters of the models

Model parameters that are adjusted in different analyses are listed in *Table 1*. The values of the synaptic weights and spatial input were chosen to allow for a large range of phase precession and stability of the network activity. For the analyses including the DG layer, weights are adjusted to allow coexistence of extrinsic and intrinsic sequences.

### Cross-correlation analysis

Cross-correlation represents the probability that a spike of one place cell would occur following a certain time lag from the spike of the another cell. Cross-correlation is always empirically computed as a histogram of time lags between spike pairs with a resolution of 5ms in a window of 200ms. Throughout the present study, the direction of a time lag is designated as the lag of the first encountered cell relative to the next cell along the trajectory, except in *Figure 1*, where the direction of time lag follows the cell order along the 0°, 45°, and 90° trajectory in each comparison group, and in *Figure 2*, where the time lag direction is from left to right cells.

Correlation lag is derived by band-pass (4–12 Hz) filtering the cross-correlation histogram and applying a Hilbert transform on the filtered signal. The phase of the analytic signal at time lag 0 is the correlation lag.

### Extrinsicity and intrinsicity

We apply quantitative measures for the extrinsic or intrinsic nature of cross-correlations in a pair of place fields following *Yiu et al., 2022*. We compare the cross-correlation histograms of a field pair for a running direction along the DG loop ($\theta_{DG}$) and opposite to the loop ($\theta_{DG} + 180°$). Extrinsicity (Ex) is computed as the Pearson's correlation ($r$) between two cross-correlation histograms, and intrinsicity (In) between the histogram of $\theta_{DG}$ and the horizontally flipped histogram of $\theta_{DG} + 180°$. The Pearson's correlation is then transformed ($r' = (r + 1)/2$) to be in the range of 0 and 1. An extrinsic correlation would give an extrinsicity near 1, since the effect of DG loop is minimal and correlation histograms are similar in both $\theta_{DG}$ and $\theta_{DG} + 180°$ directions. A pair of place fields with intrinsic correlation would see cross-correlation horizontally flipped in the $\theta_{DG} + 180°$ condition due to the large effect of DG loop, and hence, give an intrinsicity near 1. We classify a pair as extrinsic if its extrinsicity exceeds intrinsicity, and vice versa.

### Tempotron

A tempotron is a neuronally inspired classifier (readout neuron) whose dendritic synaptic weights can be adapted to recognize temporal patterns of spikes arriving at the afferents (for details, see *Gütig and Sompolinsky, 2006*). Briefly, the soma potential of the tempotron follows the equations

$$V(t) = \sum_i w_i \sum_{i,f} K(t - t_i^{(f)})$$

$$K(t - t_i^{(f)}) = V_0(\exp[-(t - t_i^{(f)})/\tau] - \exp[-(t - t_i^{(f)})/\tau_r]) \, ,$$

where $w_i$ is the adaptable weight of the afferent fiber conveying spikes from place cell $i$ to the tempotron. $K(t - t_i^{(f)})$ is a post-synaptic potential (PSP) kernel with decay and rising time constants of $\tau = 5$ ms and $\tau_r = 1.25$ ms respectively. $V_0$ is a factor which normalizes the PSP kernel to 1. A spike is said to occur if $V(t)$ crosses the firing threshold $V_\Theta = 2$ from below. After threshold crossing, the afferents will be shunted and spike arrivals will not evoke more PSPs for the rest of the pattern. A pattern is defined as the set of spike times of all the pre-synaptic place cells in a theta cycle (100 ms).

The weight $w_i$ follows the update rule

$$\Delta w_i = 0.01 \sum_{t_i^{(f)} < t_{\max}} K\left(t_{\max} - t_i^{(f)}\right)$$

$$w_i \leftarrow w_i + \Delta w_i \qquad \text{If a (+) pattern does not elicit a spike,}$$

$$w_i \leftarrow w_i - \Delta w_i \qquad \text{If a (−) pattern does not elicit a spike,}$$

where $t_{max}$ is the time at the peak of the soma potential $V(t)$. The learning rule assigns credit to the afferents based on spike timing. Spike times closer to the peak are considered to have higher contribution to the tempotron firing, hence their afferents are incremented by a larger step. After training, spike times with similar temporal correlations as the (+) patterns would be able to evoke enough PSP in the tempotron's soma and elicit a spike as a positive response of binary classification, while those similar to (-) patterns would not elicit a spike from the tempotron.

We trained the tempotrons to identify the spike patterns of place cells at locations with and without intrinsic connectivity separately. To this end, we modified our network such that DG loops are present at the upper half of the arena, spanning the space from x=-20cm to x=+20 cm at y=+20 cm in direction $\theta_{DG} = 0°$, while the loop is absent in the lower half of the arena.

During training, we applied 'non-moving' spatial inputs to the CA3 place cells at the with-loop (0 cm, 20 cm) and no-loop (0 cm, –20 cm) locations for 1 s, as if the animal were standing still at the locations, evoking the activities representing the two location cues. For computational efficiency, we restricted our analysis to the populations of CA3 place cells within the 20 cm squared boxes centered at the two locations. Each population contains 400 pre-synaptic cells, forming the input space for the tempotron. The spikes from the with-loop population will train the first tempotron and those from the no-loop population will train the second tempotron. Prior to training, the input spikes are sub-divided to 10 patterns based on their theta cycles. Each pattern has a window of 100 ms. We added noise to the patterns by jittering the spikes with Gaussian noise $\mathcal{N} \sim (0, (2\,\text{ms})^2)$ for 100 times. As a result, each tempotron receives 10×100 = 1,000 training patterns from the activity evoked by the location. All training patterns are (+) patterns and there is no (-) pattern.

After training, trajectories (20 cm long, 1 s duration) with running directions from 0° to 360° with 15° increment were simulated to cross each of the locations. The trajectories produce a mix of extrinsic and intrinsic sequences in the with-loop population and only extrinsic sequences in the no-loop population. The patterns evoked by the running trajectories were separately applied to the tempotrons. The input spikes for testing were also subdivided into theta cycles and jittered in the same manner as during training, forming 1000 testing patterns for each running direction. A sequence is said to be correctly identified if the tempotron fires at at least 1 out of 10 theta cycles along the trajectory. The accuracy rate for each running direction of trajectory is computed across the 100 jittered realizations.

## Code availability

We used Python 3 for simulations and visualization. The codes are available from a github repository (https://github.com/yyhhoi/directionalnet, copy archived at *Yiu, 2023*).

## Additional information

### Funding

| Funder | Grant reference number | Author |
| --- | --- | --- |
| Deutsche Forschungsgemeinschaft | LE2250/13-1 | Christian Leibold |
| Deutsche Forschungsgemeinschaft | INST 39/963-1 FUGG | Christian Leibold |

The funders had no role in study design, data collection and interpretation, or the decision to submit the work for publication.

### Author contributions

Yuk-Hoi Yiu, Conceptualization, Data curation, Software, Writing - original draft, Writing - review and editing; Christian Leibold, Conceptualization, Supervision, Funding acquisition, Writing - original draft, Writing - review and editing

### Author ORCIDs
Yuk-Hoi Yiu http://orcid.org/0000-0002-1997-9277
Christian Leibold http://orcid.org/0000-0002-4859-8000

Reviewer #1 (Public Review): https://doi.org/10.7554/eLife.86837.4.sa1
Reviewer #2 (Public Review): https://doi.org/10.7554/eLife.86837.4.sa2
Author Response https://doi.org/10.7554/eLife.86837.4.sa3

## Additional files

### Supplementary files
• MDAR checklist

### Data availability
The current manuscript is a computational study, so no data have been generated for this manuscript. We used Python 3 for simulations and visualization. The codes are available from https://github.com/yyhhoi/directionalnet (copy archived at *Yiu, 2023*).

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
