## [Editor Report · eLife assessment]

This **important** work presents an interesting perspective for the generation and interpretation of phase precession in the hippocampal formation. Through numerical simulations and comparison to experiments, the study provides a **convincing** theoretical framework explaining the segregation of sequences reflecting navigation and sequences reflecting internal dynamics in the DG-CA3 loop. This study will be of interest for researchers in the spatial navigation and computational neuroscience fields.

---

## [Referee Report · Reviewer #1 (Public Review)]

In the manuscript entitled "A theory of hippocampal theta correlations", the authors propose a new mechanism for phase precession and theta-time scale generation, as well as their interpretation in terms of navigation and neural coding. The authors propose the existence of extrinsic and intrinsic sequences during exploration, which may have complementary functions. These two types of sequences depend on external input and network interactions, but differ on the extent to which they depend on movement direction. Moreover, the authors propose a novel interpretation for intrinsic sequences, namely to signal a landmark cue that is independent of direction of traversal. Finally, a readout neuron can be trained to distinguish extrinsic from intrinsic sequences.

The study puts forward novel computational ideas related to neural coding, partly based on previous work from the authors, including published (Leibold, 2020, Yiu et al., 2022) and unpublished (Ahmedi et al., 2022. bioRxiv) work. The manuscript will contribute to the understanding of the mechanisms behind phase precession, as well as to how we interpret hippocampal temporal coding for navigation and memory.

---

## [Referee Report · Reviewer #2 (Public Review)]

Place cells fire sequentially during hippocampal theta oscillations, forming a spatial representation of behavioral experiences in a temporally-compressed manner. The firing sequences during theta cycles are widely considered as essential assemblies for learning, memory, and planning. Many theoretical studies have investigated the mechanism of hippocampal theta firing sequences; however, they are either entirely extrinsic or intrinsic. In other words, they attribute the theta sequences to external sensorimotor drives or focus exclusively on the inherent firing patterns facilitated by the recurrent network architectures. Both types of theories are inadequate for explaining the complexity of the phenomena, particularly considering the observations in a previous paper by the authors: theta sequences independent of animal movement trajectories may occur simultaneously with sensorimotor inputs (Yiu et al., 2022).

In this manuscript, the authors concentrate on the CA3 area of the hippocampus and develop a model that accounts for both mechanisms. Specifically, the model generates extrinsic sequences through the short-term facilitation of CA3 cell activities, and intrinsic sequences via recurrent projections from the dentate gyrus. The model demonstrates how the phase precession of place cells in theta sequences is modulated by running direction and the recurrent DG-CA3 network architecture. To evaluate the extent to which firing sequences are induced by sensorimotor inputs and recurrent network architecture, the authors use the Pearson correlation coefficient to measure the "intrinsicity" and "extrinsicity" of spike pairs in their simulations.

I find this research topic to be both important and interesting, and I appreciate the clarity of the paper. The idea of combining intrinsic and extrinsic mechanisms for theta sequences is novel, and the model effectively incorporates two crucial phenomena: phase precession and directionality of theta sequences. I particularly commend the authors' efforts to integrate previous theories into their model and conduct a systematic comparison. This is exactly what our community needs: not only the development of new models, but also understanding the critical relationships between different models.

---

## [Author Response]

The following is the authors’ response to the previous review

In response to the additional concerns voiced by Reviewer# 2, we have conducted control simulations. The outcomes are summarized in the new supplements to Figure 3. They show that the model is robust under changes of short-term plasticity parameters and running speed.

Below, we give a point-by-point response to the remaining comments of the editors and reviewers.

Editorial Assessment: This important work presents an interesting perspective for the generation and interpretation of phase precession in the hippocampal formation. Through numerical simulations and comparison to experiments, the study provides a convincing theoretical framework explaining the segregation of sequences reflecting navigation and sequences reflecting internal dynamics in the DG-CA3 loop. This study will be of interest for researchers in the spatial navigation and computational neuroscience fields.

We would like to thank the Editors very much for this positive assessment of our work!

**Reviewer #1**
In the manuscript entitled ”A theory of hippocampal theta correlations”, the authors propose a new mechanism for phase precession and theta-time scale generation, as well as their interpretation in terms of navigation and neural coding. The authors propose the existence of extrinsic and intrinsic sequences during exploration, which may have complementary functions. These two types of sequences depend on external input and network interactions, but differ on the extent to which they depend on movement direction. Moreover, the authors propose a novel interpretation for intrinsic sequences, namely to signal a landmark cue that is independent of direction of traversal. Finally, a readout neuron can be trained to distinguish extrinsic from intrinsic sequences.The study puts forward novel computational ideas related to neural coding, partly based on previous work from the authors, including published (Leibold, 2020, Yiu et al., 2022) and unpublished (Ahmedi et al., 2022. bioRxiv) work. The manuscript will contribute to the understanding of the mechanisms behind phase precession, as well as to how we interpret hippocampal temporal coding for navigation and memory.I am very pleased to have seen major improvements in the manuscript regarding (i) a clarification of the concepts of extrinsic and intrinsic mechanisms, and (ii) overall arrangement of Figures but also (iii) expanding on some important concepts such as the role of experience in determining the asymmetric connectivity that is necessary for intrinsic models of sequence generation.

We are delighted to have been able to amend the Reviewer’s concerns voiced after the initial submission. We are very grateful for their many good suggestions that allowed us to make important additions to the revised manuscript.

**Reviewer #2**
Place cells fire sequentially during hippocampal theta oscillations, forming a spatial representation of behavioral experiences in a temporally-compressed manner. The firing sequences during theta cycles are widely considered as essential assemblies for learning, memory, and planning. Many theoretical studies have investigated the mechanism of hippocampal theta firing sequences; however, they are either entirely extrinsic or intrinsic. In other words, they attribute the theta sequences to external sensorimotor drives or focus exclusively on the inherent firing patterns facilitated by the recurrent network architectures. Both types of theories are inadequate for explaining the complexity of the phenomena, particularly considering the observations in a previous paper by the authors: theta sequences independent of animal movement trajectories may occur simultaneously with sensorimotor inputs (Yiu et al., 2022).In this manuscript, the authors concentrate on the CA3 area of the hippocampus and develop a model that accounts for both mechanisms. Specifically, the model generates extrinsic sequences through the short-term facilitation of CA3 cell activities, and intrinsic sequences via recurrent projections from the dentate gyrus. The model demonstrates how the phase precession of place cells in theta sequences is modulated by running direction and the recurrent DG-CA3 network architecture. To evaluate the extent to which firing sequences are induced by sensorimotor inputs and recurrent network architecture, the authors use the Pearson correlation coefficient to measure the ”intrinsicity” and ”extrinsicity” of spike pairs in their simulations.I find this research topic to be both important and interesting, and I appreciate the clarity of the paper. The idea of combining intrinsic and extrinsic mechanisms for theta sequences is novel, and the model effectively incorporates two crucial phenomena: phase precession and directionality of theta sequences. I particularly commend the authors’ efforts to integrate previous theories into their model and conduct a systematic comparison. This is exactly what our community needs: not only the development of new models, but also understanding the critical relationships between different models.

We also would like to express our gratitude to Reviewer 2 for their numerous constructive criticisms that led to a very much improved revised manuscript!

**Reviewer #2**
1. The choice of timescale parameters for input facilitation and synaptic depression is still not fully justified in my opinion. The authors themselves mention that previous experiments suggest wide ranges for both timescales. Given that the generation of intrinsic and extrinsic sequences in their model is primarily driven by these two mechanisms, their chosen timescales should significantly impact the simulation results. I urge the authors to discuss the potential effects of selecting different sets of timescales and the possible limitations of the current selection of 500ms for both.For instance, the authors state in the caption of Fig 1 that all simulated rat trajectories were set at a speed of 20cm/s, which is a rat’s walking speed. However, the running speed of rats can exceed 3m/s. In this case, none of the CA3 cells in the model would produce any extrinsic sequences since the animal would traverse the place fields much more rapidly, preventing the sensorimotor input from increasing as it does in the model.

The reviewer raised the valid point that our simulations may be sensitive to the short-term plasticity time constants and running speeds. We therefore conducted new simulations illustrated in Figure 3—figure supplements 1 and 2.

In agreement with the reviewer’s assertion, using the current model parameters, a higher running speed would not elicit extrinsic sequences due to the lack of depolarization from spatial input (Figure 3—figure supplement 2A). However, an increase of running speed also requires sensory inputs to be available on a larger spatial scale (width of the spatial input box in our case). Parra-Barrero et al., eLife 10:e70296 and Parra-Barrero & Cheng 2023, PLOS Comput. Bio. 19:e1011101, e.g., showed that place field sizes become larger under higher running speeds and consequently lengthen the theta sequences. With such modification, along with a longer DG projection length (|r|), we were able to recover the theta sequence at a higher speed (100 cm/s), using the same STD and STF time constants (Figure 3—figure supplement 2B). Furthermore, it has been shown that theta frequency increases with running speed (e.g., Rivas et al., 1996, Exp Brain Res 108:113-8). In our analysis, a higher theta frequency (12Hz instead of 10Hz) is also able to counteract the effect of running speed and leads to control-level like phase precession (Figure 3—figure supplement 2C).

Consistent with this finding, the original study of Romani & Tsodyks 2015, Hippocampus 25:94-105, found a fourfold increase of speed (from 0.05 to 0.2 fraction of the track per second) to not affect phase-position relations (with UD = 0.8 and 800ms STD time constant), likely due to the large place field sizes covering 1/3 of the track. Thus, phase precession may only be affected by high speeds in narrow place fields in which activity would only be present for few theta cycles thus naturally having limited capacity for phase coding.

We further refrain from increasing the running speed beyond 1m/s (e.g. 3m/s as suggested by the reviewer), as the typical running speed of a rat in an 80cm square environment is between 20-40cm/s (Mankin et al. 2012, PNAS, 109:19462-19467). Even on linear tracks, reported running speeds hardly exceed 120 cm/s (e.g. Ahmed and Mehta, 2018, J Neurosci 32:73737383; Schmidt et al., 2009 J Neurosci 29:13232-13241). To our knowledge phase precession for speeds above 1.2 m/s has not been reported so far at all, certainly also owing to experimental challenges. We, however, would speculate that beyond 120 cm/s phase precession could be meaningful in large environmental enclosures with wide place cells. Thus a version of our input model with very large place field sizes should generally be able to also cover very high running speeds.

To conclude, STD and STF time constants do not need to be in a precise range to accommodate the behavioural time scales if the sensory input changes on accordingly larger spatial scales.

Following up on the reviewer’s additional concern, we also checked the effect of time constants on the theta sequences (while keeping the running speed unchanged). Decreasing the time constant of STF (τF) to 100ms would degrade the theta sequence due to a lack of depolarization, as sensory input reverts to its resting value (=0) too fast, but at 250ms, the temporal correlation of theta sequences is largely maintained (Figure 3—figure supplement 1A). However, such effects can be compensated for by an increase in sensory input which promotes input facilitation (Figure 3—figure supplement 1B). Further increasing τF does not significantly affect theta sequences as the sensory input amplitude have asymptotically reached their target values (Figure 3—figure supplement 1A bottom). The temporal correlation of theta sequences is not sensitive to the change in the time constant of STD (τD) (Figure 3—figure supplement 1C), possibly because the synaptic resource of the place cells behind the animal is reliably depleted by strong depolarization despite a fast recovery time (τD=100ms).

Since the relation between running speed and theta sequences has been thoroughly studied in Parra-Barrero et al. 2021 and Parra-Barrero & Cheng 2023, and the precise range of STD and STF time scales does not play a critical role in the temporal structure of theta sequence, we refrain from substantially revising the manuscript and only briefly add these points after Figure 3.

1. This is a point I overlooked in the initial review. The synaptic depression fraction UD is set at 0.9 or 0.7, implying that the synaptic coupling weight between CA3 excitatory cells (and CA3 to DG) is almost entirely depleted within a few hundred milliseconds. To my limited neuroscience knowledge, I am not aware of any experimental results that corroborate this potentially bold setup, and I urge the authors to provide relevant experimental and theoretical references if they exist.Most crucially, I find this setup biased towards supporting the authors’ theory for intrinsic sequences because it essentially eliminates the possibility of any CA3 cell producing an effective output to other neurons after it fires. Hence, I question whether the simulation results would be much less clean if a more moderate depression factor UD were utilized.

We thank the reviewer for giving us the opportunity to further clarify. (1) Probabilties of synaptic release (here called UD for consistency with the original work by Romani and Tsodyks), can attain a very wide range and indeed achieve values up to 0.9 (for review see e.g. Dobrunz LE, Stevens CF, 1997, Neuron, 18: 995-1008). (2) Contrary to the reviewer’s impression, a higher UD (0.7-0.9 in our case) would bias the simulation towards even more extrinsicity. Larger UD produces steeper phase precession in extrinsic sequences, because it (temporarily) generates an even stronger asymmetrical connectivity. (3) The extreme value of 0.9 was only used in Figure 1 to best illustrate the original Romani and Tsodyks 2015 idea. (4) Our simulations without recurrent synaptic connections (Figure 6) do not even require short-term synaptic depression. In view of these arguments we refrained from making further additions to the paper and refer the critical reader to this comment.

I have a few final suggestions for the authors in the hopes of further improving the manuscript for the neuroscience community:line 62: sensorimotor input is present or ABSENT?

Intrinsic activity signatures are found ”EVEN when sensorimotor feedback is present”, as one may assume that this input may be able to completely override the intrinsic patterns.

line 76: played out. colloquial, consider rewriting/explaining

We use ”evoked” now.

line 104: second part of motivation for Izhikevich-type model is wordy, and grammatically incorrect.

We have shortened the sentence.

on potential limitations of the model lines 116-120: is the use of a box an important assumption, as opposed to a more graded function, exponential or gaussian?

Using-spike based input, it is not straight-forward how to implement a graded input. One way would be to employ a stochastic point process with graded firing probability. We, however, chose to use a nonlinear facilitation function (see below).

line 124 (equation) and 129-130: How crucial is the non-linearity in the synaptic variable for the results? This is a strong assumption, as the nonlinearity is the dominant effect (as opposed to a correction/perturbation). Are there any other contributions for this ramp of activity due to sensory input?

We found results to fit best with a non-linear facilitation function (see above), and, as argued in the manuscript, facilitation indeed acts non-linearly owing to the calcium-dependence of synaptic release. We have added a comment to the Methods section explaining that we use facilitation to generate a graded spatial input.

line 187: ’...neglecting gamma activity in the model.’ I suggest removing this part of the sentence, unless you motivate why gamma would be relevant and the conditions for its generation.

We have followed the reviewer’s suggestion.